# Pharmacologically controlling protein-protein interactions through epichaperomes for therapeutic vulnerability in cancer

Suhasini Joshi[1,18], Erica DaGama Gomes[1,18], Tai Wang[1,18], Adriana Corben[2], Tony Taldone[1], Srinivasa Gandu [1], Chao Xu[1], Sahil Sharma [1], Salma Buddaseth[1], Pengrong Yan [1], Lon Yin L. Chan[1], Askan Gokce[2], Vinagolu K. Rajasekhar[3], Lisa Shrestha[1], Palak Panchal[1], Justina Almodovar[1], Chander S. Digwal [1], Anna Rodina [1], Swathi Merugu[1], NagaVaraKishore Pillarsetty [4], Vlad Miclea [5], Radu I. Peter [5], Wanyan Wang [1], Stephen D. Ginsberg[6,7], Laura Tang[2], Marissa Mattar[8], Elisa de Stanchina[8,9], Kenneth H. Yu[10], Maeve Lowery[10], Olivera Grbovic-Huezo [10], Eileen M. O'Reilly [10], Yelena Janjigian[11], John H. Healey [3], William R. Jarnagin[3], Peter J. Allen[3,12], Chris Sander[13,14], Hediye Erdjument-Bromage [15,16], Thomas A. Neubert[15,16], Steven D. Leach [10,17] & Gabriela Chiosis [1,11✉]

Cancer cell plasticity due to the dynamic architecture of interactome networks provides a vexing outlet for therapy evasion. Here, through chemical biology approaches for systems level exploration of protein connectivity changes applied to pancreatic cancer cell lines, patient biospecimens, and cell- and patient-derived xenografts in mice, we demonstrate interactomes can be re-engineered for vulnerability. By manipulating epichaperomes pharmacologically, we control and anticipate how thousands of proteins interact in real-time within tumours. Further, we can essentially force tumours into interactome hyperconnectivity and maximal protein-protein interaction capacity, a state whereby no rebound pathways can be deployed and where alternative signalling is supressed. This approach therefore primes interactomes to enhance vulnerability and improve treatment efficacy, enabling therapeutics with traditionally poor performance to become highly efficacious. These findings provide proof-of-principle for a paradigm to overcome drug resistance through pharmacologic manipulation of proteome-wide protein-protein interaction networks.

[1] Chemical Biology Program, Memorial Sloan Kettering Cancer Center, New York, NY 10065, USA. [2] Department of Pathology, Memorial Sloan Kettering Cancer Center, New York, NY 10065, USA. [3] Department of Surgery, Memorial Sloan Kettering Cancer Center, New York, NY 10065, USA. [4] Department of Radiology, Memorial Sloan Kettering Cancer Center, New York, NY 10065, USA. [5] Faculty of Automation and Computer Science, Technical University of Cluj-Napoca, Cluj-Napoca, CJ 400114, Romania. [6] Center for Dementia Research, Nathan Kline Institute, Orangeburg, NY 10962, USA. [7] Departments of Psychiatry, Neuroscience & Physiology, and the NYU Neuroscience Institute, New York University Grossman School of Medicine, New York, NY 10016, USA. [8] Antitumour Assessment Core Facility, Memorial Sloan Kettering Cancer Center, New York, NY 10065, USA. [9] Molecular Pharmacology Program, Memorial Sloan Kettering Cancer Center, New York, NY 10065, USA. [10] David M. Rubenstein Center for Pancreatic Cancer Research, Memorial Sloan Kettering Cancer Center, New York, NY 10065, USA. [11] Department of Medicine, Memorial Sloan Kettering Cancer Center and Weill Cornell Medical College, New York, NY 10065, USA. [12] Department of Surgery, Duke University School of Medicine, Durham, NC 27710, USA. [13] Department of Data Science, Dana-Farber Cancer Institute, Boston, MA 02115, USA. [14] Department of Cell Biology, Harvard Medical School, Boston, MA 02115, USA. [15] Department of Cell Biology, New York University Grossman School of Medicine, New York, NY 10016, USA. [16] Kimmel Center for Biology and Medicine at the Skirball Institute, NYU School of Medicine, New York, NY 10016, USA. [17] Dartmouth Geisel School of Medicine and Norris Cotton Cancer Center, Lebanon, NH 03766, USA. [18] These authors contributed equally: Suhasini Joshi, Erica DaGama Gomes, Tai Wang. ✉email: chiosisg@mskcc.org

Proteins function not in isolation but as components of intricate networks of macromolecules linked through interactions represented as 'interactomes' (i.e., protein-protein interaction (PPI) networks)[1–3]. In this context, disease states reflect disruptions in the intrinsic tissue-specific and cell-specific interactomes that arise from both the internal and external stressors[4]. Disease interactomes are thus maps of alterations induced by these stressors in intracellular protein connectivity and in turn, within the system as a whole[1–3]. Understanding how such complex disease interactomes can be controlled for therapeutic efficacy is an important and understudied topic in discovery science.

Several elegant concepts and computational frameworks have been developed to address the controllability of biological systems[5–7], which in cancer cells may equate to directing them from a drug-resistant state to a state of vulnerability. Applied to 'omics datasets, these approaches helped identify key control genes and potential drug targets in cancer[8–11]. The inherently dynamic nature of interactome networks, indicating the need to adapt and reflecting the system's ability to respond to changes from external cues and conditions, poses however a challenge to such discovery methods that rely on 'omics datasets representing a descriptive inventory of biomolecules or that measure changes in their stoichiometry at a given time and condition[12]. This is especially true in cancer, where cells, and in turn interactomes, continuously evolve to adapt to challenging microenvironments and to withstand therapeutic assaults[13,14].

Large-scale interactome changes that occur in disease, where stressors of epigenetic, proteotoxic, and environmental nature combine with genetic defects to remodel how proteins interact and assemble[1,15], require a backbone whereby alterations in connectivity are promoted and maintained. Support is provided by the structural and functional restructuring of the chaperome, an assembly of over 300 chaperones, co-chaperones and other factors[16,17], into maladaptive long-lived oligomeric structures termed epichaperomes[18–23]. Not to be confused with chaperones, ubiquitous proteins which fold and act through one-on-one dynamic complexes, epichaperomes act as pathologic scaffolds that form specifically in disease[18–23]. They cause thousands of proteins to improperly interact and organise inside cells. HSP90 and HSC70 chaperones play a central role in the formation of epichaperome structures, yet these chaperones become distinct entities when part of epichaperomes[18–24]. These features provide an opportunity for small molecules that specifically target the epichaperome conformation to discriminate from the more abundant chaperone proteins[18,20,22,24]. Unlike ubiquitous chaperones, epichaperomes are exclusively localised to diseased cells and tissues[18–24].

There is a direct, causal link between epichaperome formation and interactome changes in disease, with interactome connectivity and functional imbalances reverting upon epichaperome disassembly into normal chaperome components[22,25,26]. In cancer, maintenance of the epichaperome-mediated interactome network is vital for cancer cell survival[18,19,24,27]. The higher the epichaperome levels, the higher the number of proteins being negatively impacted, i.e., the higher of number of aberrant PPIs, and the greater the vulnerability of cancer cells to therapeutic intervention[18,21]. Independent of tissue of origin, tumour subtype, or genetic background, approximately 50–60% of tumours express variable epichaperome levels, with ~10–15% being high expressors, thus characterised by interactome hyperconnectivity (i.e., a state whereby no rebound pathways can be deployed)[28] and maximal vulnerability[18,19].

Herein, we exploit these advances in the biology of cellular stress for interactome control in disease with the tacit goal of achieving therapeutic vulnerability. We posit by understanding how epichaperomes rewire interactomes, we can modulate, and anticipate, dynamics of interactome networks critical for understanding disease biology and drug resistance. Understanding the biology of interactome plasticity, will enable engineering interactome hyperconnectivity for vulnerability. We posit forcing interactomes into a vulnerable state of hyperconnectivity, or maximal PPI network capacity, may prime cells to existing treatments. To test these hypotheses, we apply a chemical biology approach we invented for systems-level exploration of protein connectivity changes[22,29] and combine it with biochemical and functional validation studies. We choose pancreatic cancer as a representative model—including cancer cell lines, fresh patient biospecimens, and cell- and patient-derived xenografts in mice—for our case study. Interactome network plasticity in this cancer, which arises from highly redundant signalling pathways, poses a real challenge to therapy and accounts for treatment resistance[30–32].

We demonstrate our systems-level approach delivers a mechanistic understanding of how proteome-wide PPI networks can be controlled to manipulate drug sensitivity/resistance. We find interactomes can be forced into a state of hyperconnectivity through pharmacological modulation of epichaperomes, effectively shutting down the deployment of alternative signalling pathway routes. Analogous to baseline hyperconnectivity[18], pharmacologically induced hyperconnectivity is performed by increasing the number of interactions HSP90 establishes with the proteome through recruitment of HSP70 chaperones and co-chaperones. Pharmacologically induced hyperconnectivity retains intrinsic protein pathway activity and cellular phenotype by reactivating protein pathways already in existence, and constitutively active, at baseline. Whereas new connections are established at the hyperconnectivity state, the goal of these PPIs is to retain overall signalling through constitutively active pathways. Taking advantage of these mechanistic outcomes, we applied pharmacologically induced hyperconnectivity to poor-outcome Ras-driven cancers to show that current therapeutics with poor performance in this context become effective when tumours are primed into a state of interactome hyperconnectivity. Accordingly, we provide a previously unappreciated treatment strategy for cancer where the capacity of PPI networks, not genetic or epigenetic alterations, defines therapy.

## Results

**Epichaperome occupancy in pancreatic cancer.** We first investigated epichaperome abundance in pancreatic cancer as a surrogate measure of baseline interactome network integration, or PPI network connectivity status[18,28]. This analysis is based on previous findings in cancer, whereby the more of the chaperome components in a cell switch to the epichaperome, the higher the connectivity of cell-wide PPI networks, and in turn, the greater the vulnerability of the cell to inhibition of epichaperome network hubs[18,21,25,26].

We used primary specimens ($n = 39$), cell lines ($n = 11$) and patient-derived cells ($n = 3$) for this analysis. We chose the MDA-MB-468 cell line for comparison as it is a high-epichaperome, high PPI network connectivity (i.e., interactome hyperconnectivity) and high-sensitivity standard[18,33]. Specifically, we assessed the apoptotic sensitivity of these specimens to epichaperome inhibition by the small molecule PU-H71 (Fig. 1a, b) and performed correlative analyses between cell sensitivity and epichaperome levels (as measured using the PU-FITC epichaperome probe[34]) (Fig. 1b and Supplementary Fig. 1).

PU-H71 is an HSP90 inhibitor that kinetically selects for the epichaperome over HSP90 pools, as it becomes trapped when bound to HSP90 in epichaperomes, whereas it exits rapidly from

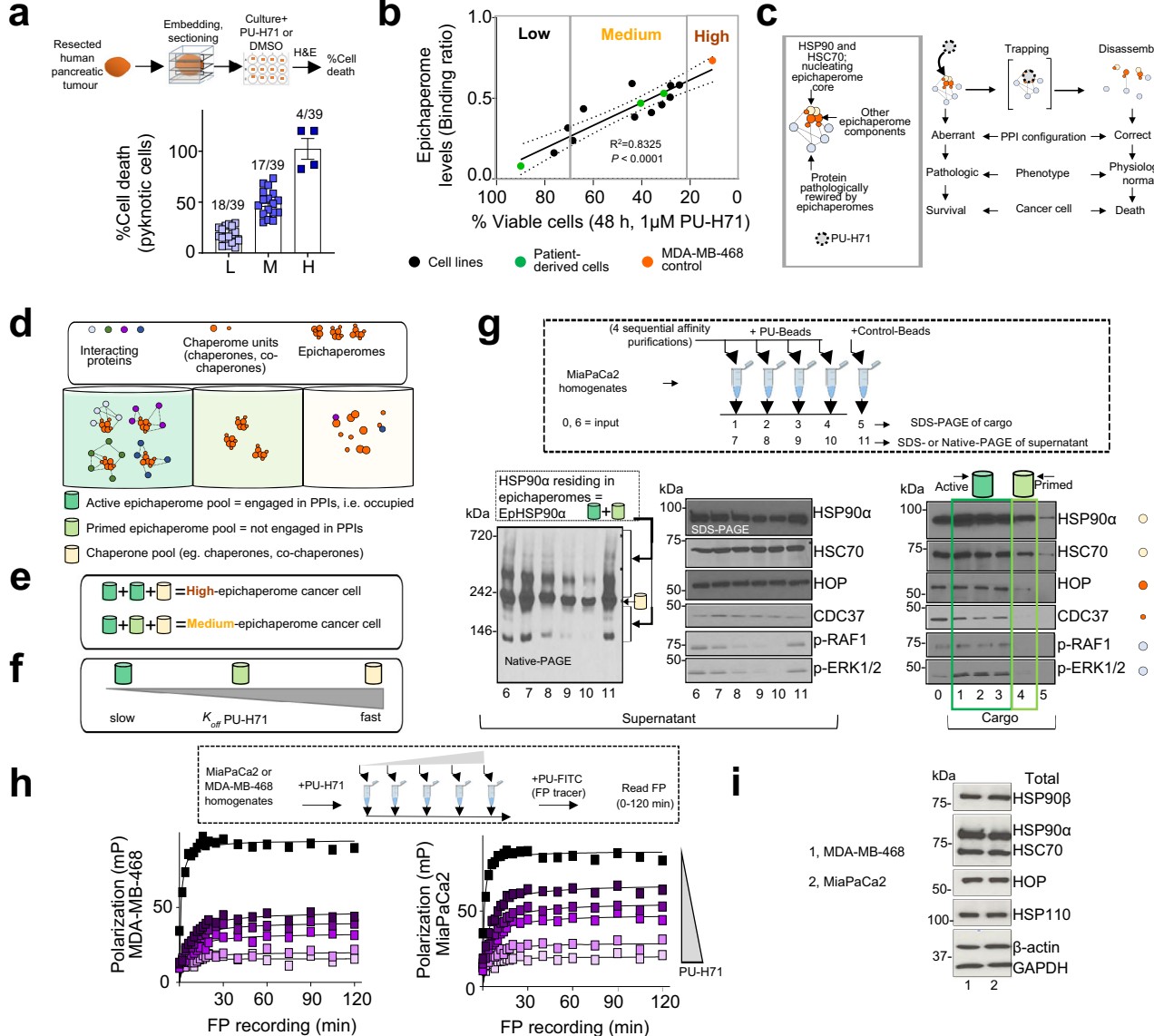

**Fig. 1 Epichaperome occupancy is associated with sensitivity to PU-H71 in pancreatic cancer cells. a** Viability of human pancreatic cancer surgical explants (n = 39) treated for 48 h with PU-H71 (1 μM). Values were normalised to DMSO (vehicle) treated explants. Error bars represent SEM. L, low; M, medium, H, high. **b** Correlative analysis performed in pancreatic cancer cells between cytotoxicity (evaluated by Annexin V staining) and epichaperome levels (see Supplementary Fig. 1). MDA-MB-468 cell line, high-sensitivity, high-epichaperome control. Each data point is the mean of triplicate experiments. Pearson's r, two-tailed, n = 14 cell types. **c** Schematic of the biochemical and functional effect of PU-H71 on cancer cells via epichaperome supression. Grey box, schematic of an epichaperome oligomer. **d** Schematic showing the heterogenous pools of chaperome assemblies in a cancer cell. **e** Composition of epichaperome pools in high- and medium-epichaperome cancer cells. **f** PU-H71's relative dissociation rate constant ($k_{off}$) for the distinct chaperome assemblies. **g** Sequential affinity purification using a solid support immobilised PU-H71 (PU-beads). Representative gels of three experimental repeats are shown for cargo and supernatant. **h** Fluorescence polarisation (FP) analysis performed in MDA-MB-468 vs MiaPaCa2 cell lysates pre-incubated with increasing concentrations of PU-H71. Each curve represents the mean of three biological replicates, and the data is representative of three independent experiments. **i** Western blot analysis of chaperones that are key components in epichaperome formation. β-actin and GAPDH, protein loading controls. Representative gels of three experimental repeats are shown.

HSP90[18,24,33]. The biochemical and functional mechanism of epichaperome modulation by PU-H71 comprises a first step of trapping of epichaperomes along with their interacting proteins[18,22,35]. This step is followed by epichaperome collapse and restoration of normal PPI network connectivity[20,22], detrimental to both the maintenance of a malignant phenotype and cancer cell survival (Fig. 1c and ref. [18,19]). Importantly, the dissociation rate constant ($K_{off}$) of PU-H71 from epichaperomes is proportional to epichaperome occupancy with a $K_{off}$ from active epichaperomes (i.e., proteome bound) < $K_{off}$ from primed

epichaperomes (i.e., not proteome bound) (Fig. 1d–f schematics and ref. [18,35]). Proteome in this context is the full complement of proteins being bound to and impacted by epichaperomes in a specific cellular context. This indicates PU-H71, when used at well-defined concentrations, can be used to preferentially engage and interrogate only active epichaperomes or both active and primed epichaperomes.

Of the 53 samples, we found 7% were of high-epichaperome, high-vulnerability to PU-H71 ( > 80% apoptosis), indicative of a cellular state with the epichaperome complement fully engaged in

interactions with the proteome (i.e., a cell containing majorly active epichaperomes), and in turn, of interactome networks devoid of redundant paths (i.e. state of interactome hyperconnectivity and maximal PPI network capacity)[18,19,28]. Conversely, 53% of tumours were medium-sensitivity tumours (30%< apoptosis <80%, Fig. 1a, b), thus tumours which comprise both active and primed epichaperomes[18]. This is evidenced by sequential capture experiments performed with a solid-support immobilised PU-H71 (i.e., PU-beads)[36] (Fig. 1g, see the sequence in the depletion of epichaperome pools by the PU-beads), by fluorescence polarisation experiments comparing the binding profile of PU-H71 to cell homogenates obtained from a high- and a medium-epichaperome cell line (Fig. 1h, i, see the tighter binding of PU-H71 to MDA-MB-468 homogenates when compared to MiaPaCa2 despite similar overall levels of HSP90 and of other epichaperome-constituent chaperones in both cell lines), and was further validated through biochemical and functional assays described below.

**Epichaperome occupancy shapes recovery.** Unused epichaperome capacity in the medium-sensitivity tumours could signify a mechanism of redundancy within PPI networks[12,28]. Network redundancy refers to the inclusion of extra components not strictly necessary to functioning but used in case of failure in other components. Redundancy is built into interactome networks to enable cellular survival in the advent of continuous fluctuations in the extra- and intra-cellular environment and to facilitate recovery from the failure of a specific component.

To test this hypothesis, we evaluated cell viability after drug wash-off, a treatment condition more likely to capture the intrinsic dynamics of interactome networks and provide the mechanisms of redundancy. We used MiaPaCa2 pancreatic cancer cells, a medium-epichaperome cell line, thus characterised by both active and primed epichaperomes, and the MDA-MB-468 control, a high-epichaperome cell line, and tested conditions where we removed the inhibitor from the cell culture after 24 h and monitored after inhibitor wash-off both epichaperome levels (Fig. 2a, $T_0$ to $T_{72}$ by Native PAGE) and cell recovery up to 72 h (Fig. 2b–d). We treated cells with PU-H71 at 1 µM, a concentration that engages mostly active epichaperomes, and with 5 µM, a concentration that engages both active and primed epichaperome pools (ref. [25,33] and see further). MDA-MB-468 was included as control for a cellular system comprised by majorly active epichaperome pools, revealing maximal PPI network capacity characterised by hyperconnected networks. It is included to show side-by-side the effect on cellular networks and cell viability in conditions of both total and partial epichaperome suppression.

At the tested PU-H71 concentration of 0.5, 1 and 5 µM, no viable cells were observed for MDA-MB-468 in the 3 days of the observation period after PU-H71's wash-off, as evidenced by crystal violet staining (Fig. 2c), ATP levels (Fig. 2d) and live-cell microscopy (Fig. 2e). Conversely, for MiaPaCa2, we observed cell recovery in inverse proportion to the used concentration of PU-H71, and thus, to epichaperome suppression by PU-H71. Complete epichaperome suppression (i.e., of both primed and active epichaperome pools) by 5 µM PU-H71, but not of partial (i.e., of mainly active epichaperome pools) by 1 µM PU-H71, effectively stopped cell recovery after drug wash-off (Fig. 2a–e).

The relationship between the drug's epichaperome suppression capacity and cell recovery was also evident for Debio032 (CUDC-305)[37], an HSP90 inhibitor structurally similar to PU-H71 yet ineffective at totally suppressing epichaperomes (Fig. 2e). Albeit tested at the maximal biologically active concentration[37], CUDC-305's inability to totally supress epichaperomes resulted in cell growth recovery after inhibitor wash-off (Fig. 2f).

Engagement of both active and primed epichaperome pools by PU-H71 (see $T_{0-72}$, MiaPaCa2 cells treated with 5 µM PU-H71 and MDA-MB-468 with either 0.5, 1 and 5 µM, Fig. 2a, e) resulted in epichaperome disassembly (see Native-PAGE), suppression of aberrant PPI network activity (e.g., see p-MEK/MEK as a read-out of MAPK signalling, and see further) and cell death (see XIAP and PARP). Conversely, in the medium-epichaperome expression MiaPaCa2 cells released from a 24 h exposure to 1 µM PU-H71 (i.e., a concentration of PU-H71 mostly engaging the active but not the primed epichaperome pools), we observed a paradoxical increase (as opposed to disassembly) in active epichaperomes to levels that neared those observed at baseline in the high-epichaperome MDA-MB-468 cells (Fig. 2a).

Biochemically therefore, cell recovery, or lack thereof, is reflected by epichaperome activity (i.e., does the tumour express both active and primed or only active epichaperomes?) and depends on the ability of a drug to supress such activity (i.e., is the concentration of drug delivered to the tumour enabling suppression of both active and primed epichaperomes?). This is supported by clinical observations, where a tumour concentration of 5 µM and higher PU-H71 recorded at 24 h post drug administration was shown to result in durable responses in epichaperome-positive poor prognosis malignancies such as metastatic triple-negative breast cancer and myeloproliferative neoplasms transformed to acute myeloid laeukemia[25,26,33].

In addition to highlighting a positive relationship between epichaperome suppression and drug efficacy, the studies above suggest a potential mechanism of rebound. Specifically, the paradoxical epichaperome increase observed in MiaPaCa2 cells after a 24 h exposure to suboptimal 1 µM PU-H71, may indicate a temporary state where the MiaPaCa2 interactome may be forced into a state of hyperconnectivity and therapeutic vulnerability while attempting to recover, a hypothesis we investigate further.

**Recovery is a state of temporary hyperconnectivity.** To gain an unbiased, proteome-wide understanding of the state of recovery, we applied a large-scale functional proteomics method we invented, termed epichaperomics or chemical chaperomics[22,29] (Fig. 3a and Supplementary Data 1). Epichaperomics investigates connectivity between the epichaperome and the proteome, provides information on the nature and number of connections (i.e., paths) established, and importantly, unlike other 'omics, it directly informs on the functional output of such connections[22,29].

Epichaperomics is an affinity purification method that uses the multitude of epichaperomes in a particular cell as bait to capture the disease-promoting complement of proteins and query their disease-specific interactions[29]. An analogy is affinity purification methods whereby several tagged proteins are added to the cell and interacting proteins are captured. Unlike classical affinity purification, in epichaperomics there is no need to introduce artificial baits because the epichaperomes themselves are the baits[29]. Also, in epichaperomics there is not one bait but rather a multitude of baits (i.e., the epichaperome oligomers characteristic of a specific cellular context), each having distinct interactors[29]. Similar to other affinity purification methods, bait-bound interactors are captured and interrogated by mass spectrometry for identification. In the case of epichaperomics, the epichaperome biological baits are captured by chemical probes that trap epichaperomes bound to their interactors (e.g., PU-beads), thus retaining interactions through subsequent isolation steps and enabling their unbiased identification by mass spectrometry[29]. Similar to other conventional affinity purification methods, epichaperomics does not indicate which proteins are directly interacting with each other. Bioinformatic inference, not direct physical experiments, is used to generate proteome-wide

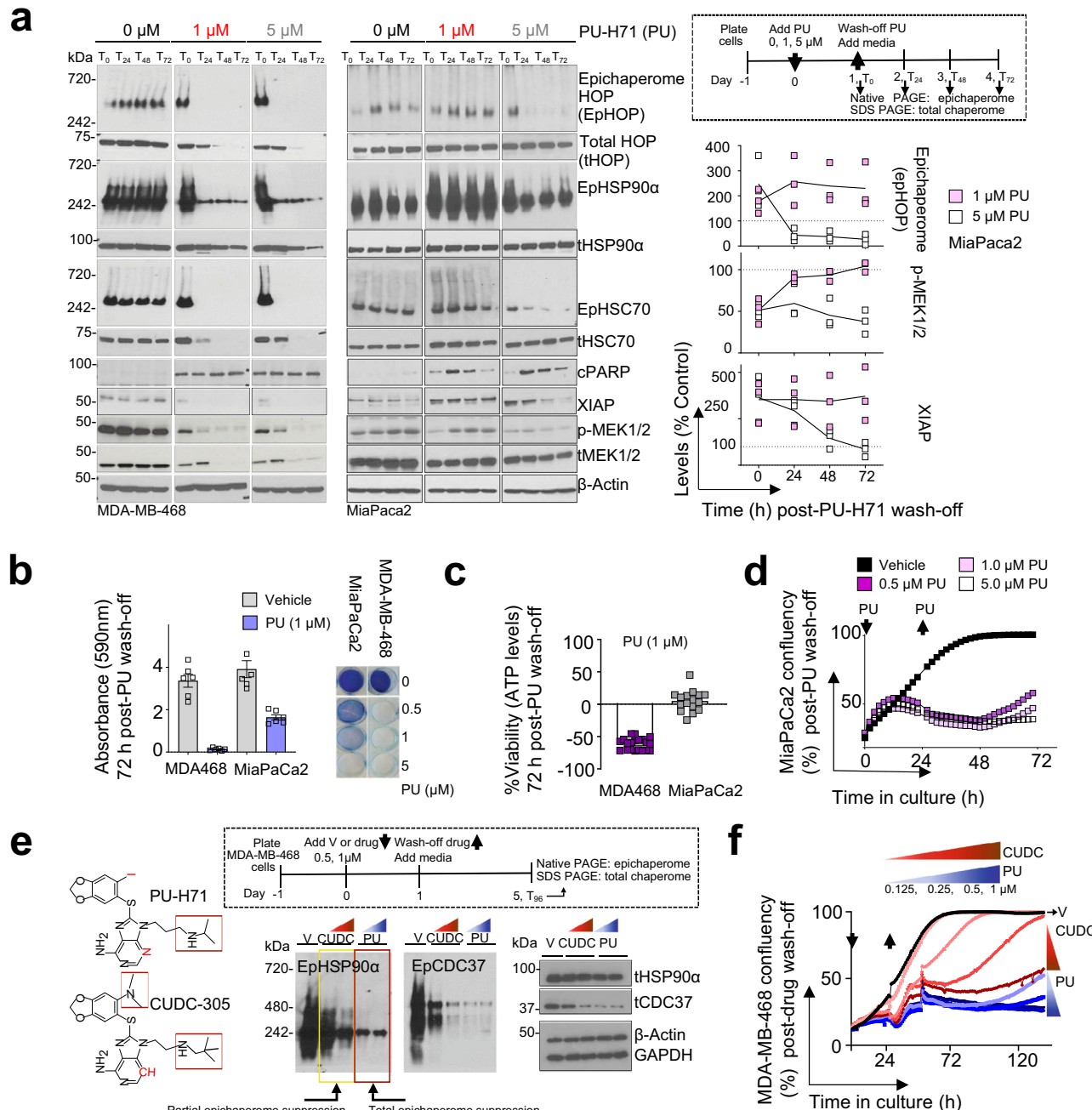

**Fig. 2 Unoccupied epichaperome capacity enables recovery. a** Dynamics of epichaperome and total chaperome pools at 0 to 72 h following treatment of cells for 24 h with PU-H71 (0, 1, 5 μM). $T_0$ to $T_{72}$, 0 h to 72 h post-inhibitor wash-off from the cell culture. Native-PAGE, epichaperome levels; SDS-PAGE, total chaperome levels; p-MEK, read-out for MAPK signalling pathway activity; XIAP and cleaved PARP, cell survival and death markers, respectively; β-actin, protein loading control. Graph, individual datapoints of 3 experimental repeats for MiaPaCa2. **b** Viability of cells at 72 h post-PU-H71 (PU) or -vehicle removal. Cells were treated for 24 h with PU or vehicle prior to wash-off and cytotoxicity evaluation. Error bars represent SEM, n = 6 biological replicates. Representative crystal violet stained cells are also shown. **c** Same as in (**b**) for cell viability assessed by an ATP-based assay (n = 16 biological replicates from 3 individual experiments). Negative values represent a decrease in cell numbers compared to the initial cell population. Error bars represent SEM. **d** Cell confluency monitored by live-cell microscopy (IncuCyte) over the interval 0 to 72 h of cells treated as in (**a**). Each curve represents the mean of 3 wells, and the data is representative of 3 independent experiments. **e** Epichaperome and total chaperome pools at 96 h following treatment of cells for 24 h with Vehicle (V), PU-H71 (0.5, 1 μM) or CUDC-305 (0.5, 1 μM). $T_{96}$, 96 h post-inhibitor wash-off from the cell culture. β-actin and GAPDH, protein loading controls. The chemical structure of PU-H71 and CUDC-305 is also shown. Red box and/or symbol indicate structural differences. The data is representative of 3 independent experiments. **f** Cell confluency monitored by live-cell microscopy (IncuCyte) over the interval 0 to 160 h of cells treated as in (**e**) with the indicated concentrations of inhibitors. Each curve represents the mean of 3 wells, and the data is representative of 3 independent experiments.

interactome maps[29]. In most cases, including for the purpose of this study, it is not important to know the direct binary interactions at the molecular or quantitative level, only which maladaptive proteins and pathways are stabilized which can be

learned from the interactome data and bioinformatics, with further evidence based on functional assays and outcome assessments, as supported by our previous publications[18,20,22,29]. To summarize, as with any affinity purification methods, PPIs

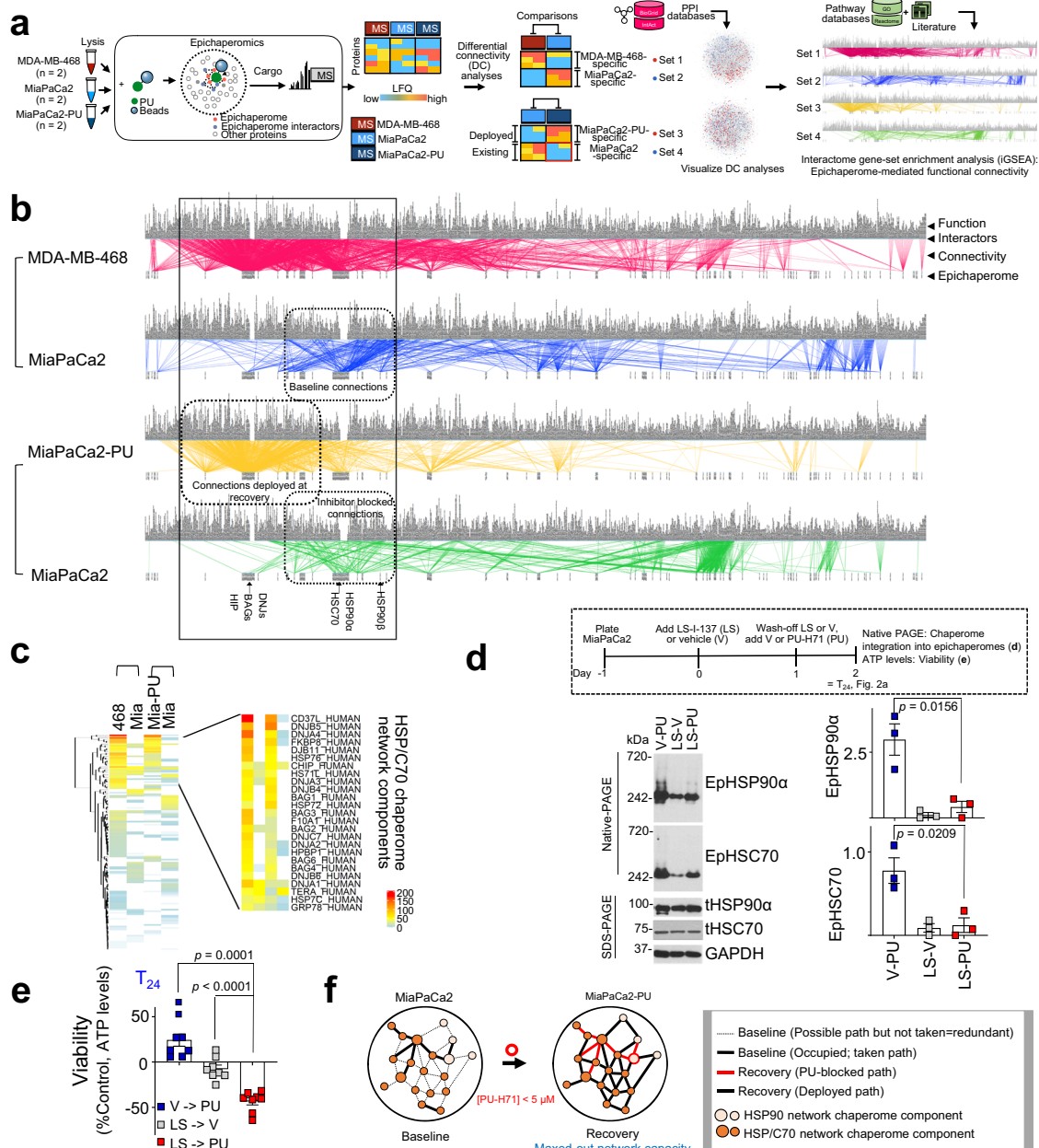

**Fig. 3 Recovery is a state of interactome hyperconnectivity and maximal network capacity executed by integration of chaperome networks. a** Schematic of systems level analyses to determine the number of connections and the functional outcome of connections established by the epichaperome and its interacting proteome. **b** Epichaperome-proteome functional connectivity maps. Each chaperome unit of the epichaperomes is listed at the bottom of the map (see label, epichaperome), paths established between the epichaperome and proteome are depicted by a line (see label, connectivity), and the functional output of each connection is listed on top (see label, function). Connectivity maps are created by differential analysis to emphasise the difference in baseline PPI networks (MDA-MB-468 vs MiaPaCa2) and between baseline and recovery states (MiaPaCa2 vs MiaPaCa2-PU). **c** Cluster analyses identify components of the HSP/C70 chaperome network as key epichaperome effectors of the bypass connections established at recovery. Scale bar, number of functional connections. **d** LS-I-137 (LS), an allosteric HSP/C70 binder, blocks HSC70 incorporation into epichaperomes and limits deployment of redundant paths. LS, 250 nM; PU, 1 µM. Each data point is an independent experiment. Error bars mean ± SEM; unpaired two-tailed t test. **e** Viability of cells treated as in (**d**). Error bars mean ± SEM; n = 8 biological replicates from 3 independent experiments, one-way ANOVA with Dunnett's post-hoc. **f** Schematic summary of the findings.

networks are generated through experimentation (i.e., physical determination of binding interactors) combined with PPI database mining.

We performed pair-wise comparisons of epichaperomics datasets from MDA-MB-468 and MiaPaCa2 to investigate the capacity of baseline interactome networks (i.e., How many connections are established between the epichaperome and the proteome in cells at baseline?), and from MiaPaCa2 and

MiaPaCa2 released from a 24 h exposure to suboptimal 1 µM PU-H71 (MiaPaCa2-PU) to understand network dynamics and the nature of the recovery mechanisms (i.e., How are recovery paths deployed? What is the nature of these paths? How do recovery paths remodel cell-wide interactome networks?). It is important to note that epichaperomics at recovery informs on newly formed connections because baseline active epichaperomes remain occupied by PU-H71 even after its removal from

the cell culture media (Figs. 2, 3, Supplementary Fig. 2 and ref. [18,22,24,25,33,35]).

We first generated interactome maps for each condition and established epichaperome-proteome functional connectivity maps specific to each paired comparison. In MDA-MB-468 cells we observed epichaperomes established 3,046 unique functional connections (Fig. 3b, red lines). In this context, we define functional connections as those established by epichaperomes with protein pathways, through one or more proteins that are part of such pathway, and therefore represent the active epichaperome pools. Conversely, only 893 connections were established in MiaPaCa2 cells (Fig. 3b, blue lines), indicative of unused epichaperome capacity (i.e. the presence of primed epichaperomes along active epichaperome pools), which was also observed experimentally (Figs. 1 and 2).

When comparing MiaPaCa2 to MiaPaCa2-PU, we found 1,502 connections unique to MiaPaCa2-PU compared to 930 connections in MiaPaCa2 (Fig. 3b, yellow and green lines, respectively). Of importance, in MiaPaCa2 cells at recovery, the combined effect of baseline paths being inhibitor blocked and new bypass paths being deployed, potentially created a temporary state of network hyperconnectivity and of maximal PPI network capacity, nearing what we observed in MDA-MB-468 cells at baseline ($\sim$930 + 1,502 = 2,432 functional connections in MiaPaCa2-PU vs 3,046 functional connections in MDA-MB-468, Fig. 3b). These systems level analyses support the notion that additional active epichaperome pools form in MiaPaCa2 cells released from a 24 h exposure to 1 μM PU-H71, as was evidenced by Native PAGE monitoring (Fig. 2a).

We next investigated the nature of the newly formed active epichaperomes. We found in MiaPaCa2-PU newly formed connections occurred through paths intrinsic to, and occupied, at baseline in MDA-MB-468 cells, but free (i.e., possible but not taken, redundant paths) in MiaPaCa2 (compare red and yellow lines in the black box, Fig. 3b). We found these newly formed paths were executed majorly by co-opting to epichaperome structures additional co-chaperones, such as those of the HSP/C70 chaperone network (Fig. 3c). We confirmed this both biochemically and functionally by showing that, pharmacologically blocking HSP70/C70 recruitment into epichaperomes prior to PU-H71 treatment, inhibited deployment of redundant paths (Fig. 3d, see epichaperome levels between V-PU and LS-PU) and rendered MiaPaCa2 cells as vulnerable to PU-H71 as those with baseline hyperconnected PPI networks (Fig. 3e).

Thus, analogous to baseline interactome hyperconnectivity, observed in MDA-MB-468 and other high-epichaperome tumours[18], pharmacologically induced hyperconnectivity is executed by enhancing the number of interactions and interaction partners between components of the two major chaperome networks, HSP90 and HSP70 (Fig. 3f). This engineered interactome state reflects a maxed-out capacity of cellular PPI networks (i.e. a state where no redundant pathways can be deployed) as judged by the number of established functional connections and its vulnerability to PU-H71.

**Rebound is encoded in baseline interactome activity**. To understand how pharmacologically induced interactome hyperconnectivity affects proteome function, especially signalling networks, we generated physical PPI network maps for each cellular state (Fig. 4a). We observed substantial changes in cellular PPI networks with 32.28% of nodes (i.e., proteins) and 68.78% of edges (i.e., connections) being affected at recovery (i.e., the state of rebound). Despite >50% of PPIs being disturbed, we observed these paths redistributed locally, meaning they switched from a node to a closely located node (MiaPaCa2-PU vs MiaPaCa2 in

Fig. 4a, see that gained nodes, in red, and lost nodes, in blue, are in close proximity). Conversely, nodes were distal when comparing node distribution in baseline MiaPaCa2 and MDA-MB-468 (Fig. 4a, red and blue nodes are far apart on the PPI map).

Most pancreatic cancer cells, including MiaPaCa2, have common activating point mutations in KRAS, leading to uncontrolled activation of downstream intracellular signalling pathways, including the RAF/MEK/ERK pathway, contributing to tumour cell proliferation and survival[31,38]. Additional downstream signalling partners of KRAS include PI3K and AKT which link growth factor receptors to the phosphorylation and activation of the serine/threonine kinase, mammalian target of rapamycin (mTOR). Activation of the STAT3 pathway is yet another hallmark associated with an inflammatory phenotype and therapy resistance[39]. MiaPaCa2 cells therefore are dependent on these pathways being constitutively on. The aberrant activity of these pathways at baseline is supported by epichaperomes (Supplementary Fig. 3a and see further).

Fundamental to network analysis is the concept that proteins involved in the same cellular processes often interact with each other[29]. The local distribution of PPIs when cellular networks remodel from baseline (MiaPaCa2) to recovery (MiaPaCa2-PU) may therefore signify that cellular rebound is executed through reactivation of protein pathways already existent, and constitutively active at baseline in MiaPaCa2 yet deployed at recovery through alternate paths (i.e., by engaging or co-opting other proteins that may execute similar functions). Systems-level analysis of signalling protein pathways intrinsically dependent on the epichaperome in MiaPaCa2 ('Existent' or 'Active at baseline') and those upregulated at recovery through the epichaperome (i.e., in MiaPaCa2-PU, see 'Deployed at recovery') support this notion (Fig. 4b, c).

We validated these findings by analysing the activity of several pathways at baseline and at recovery (see Western blot analysis of pancreatic cancer signalling pathways at $T_{24-72}$ in Figs. 2a, 4c, 5a and see changes in transcription factor expression between the baseline and the hyperconnectivity state of the MiaPaCa2 cell line in Supplementary Fig. 3b and Supplementary Data 2), by demonstrating retained vulnerability of cells at recovery to inhibitors of these signalling pathways (Fig. 5c–e) and by probing the interaction of epichaperome components with key effectors of these pathways at both the baseline and recovery (see affinity purification in Figs. 1g, 5b), as we detail below.

As a modality to confirm the outcomes of our system levels analysis, we pharmacologically explored the deployed signalling routes (Fig. 4c). Specifically, we chose the Ras-MAPK signalling to inquire on proteins utilized by MiaPaCa2 cells to maintain the activity of this pathway at rebound (see heatmap in Fig. 4c). We identified that the 'deployed' MAPK signalling output is executed through an interplay of pathways such as NFκB activation by cAMP-dependent protein kinase (PKA) and through RAF activation by Phospholipase C, gamma (PLCγ) (see mapping of 'Deployed' proteins on specific signalling networks, blue box). Gene set enrichment analysis of transcriptional factors differentially expressed between MiaPaCa2-PU and MiaPaCa2 support enhanced NFκB transcriptional activity at recovery (Supplementary Fig. 3b). We then used pharmacologic agents targeting key proteins in the deployed state. These are reported to act on inhibiting NFκB transcription (JSH-23), PLC activity (U-73122) and PKA activity (PKA inhibitor fragment (6-22) amide and the small molecule H-89)[40–43]. Where available, structurally similar inactive controls were also included (eg. U-73343). SCH772984, an ERK1/2 inhibitor was also included as control to confirm ERK activation, as a read-out for Ras-MAPK signalling activity in the deployed state. We found these pharmacologic inhibitors lowered signalling output in the deployed state as evidenced by a

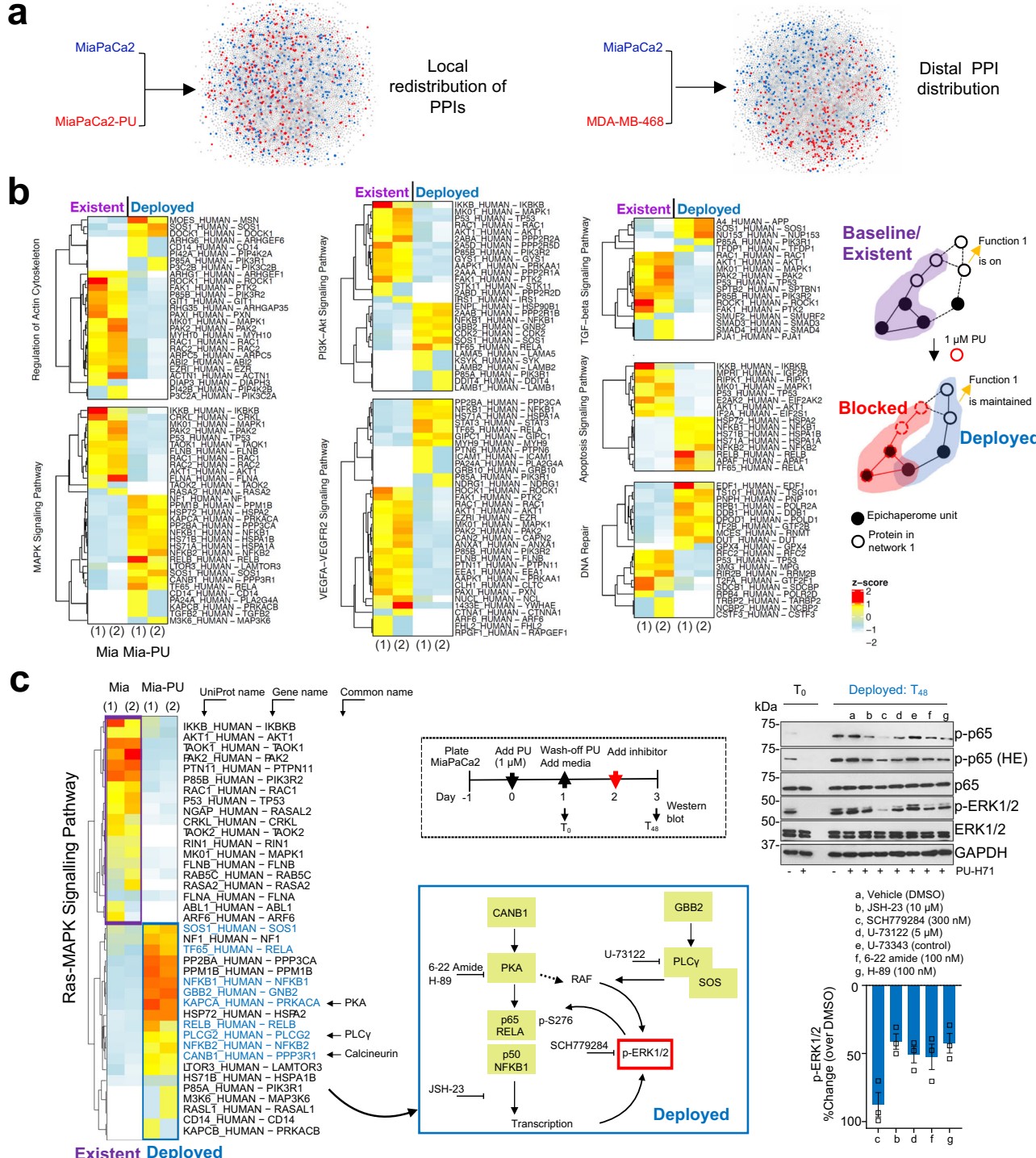

**Fig. 4 Cells forced into a state of interactome hyperconnectivity retain the activity of protein pathways already existent, and constitutively active, at baseline. a** Comparative analysis of PPI distribution. Left side map: blue, indicating proteins that establish interactions unique to baseline (MiaPaCa2) and red, unique to recovery (MiaPaCa2-PU). Right side map: red, unique to MDA-MB-468 vs blue, unique to MiaPaCa2. The node size and node colour represent the log10 transformed p-values and the fold change values, respectively, from the DC analysis (see Fig. 3a). Network edges and node labels were omitted for figure clarity. **b** Heatmaps showing proteins of key pathways activated in pancreatic cancer that interact with epichaperomes at baseline (existent, MiaPaCa2) and at recovery (deployed, MiaPaCa2-PU), as determined by epichaperomics. Each column is an experimental repeat. See Fig. 3a for experimental design. The right-side schematic shows the mechanism by which epichaperomes remodel PPIs to maintain, in conditions of interactome hyperconnectivity, the activity of protein pathways constitutively active at baseline. **c** Proof-of-principle validation of alternate paths deployed to retain a hyperactive ERK in conditions of induced network hyperconnectivity. Drugs were added as indicated in the schematics. Error bars mean ± SEM of $n = 3$ independent experiments. HE high exposure.

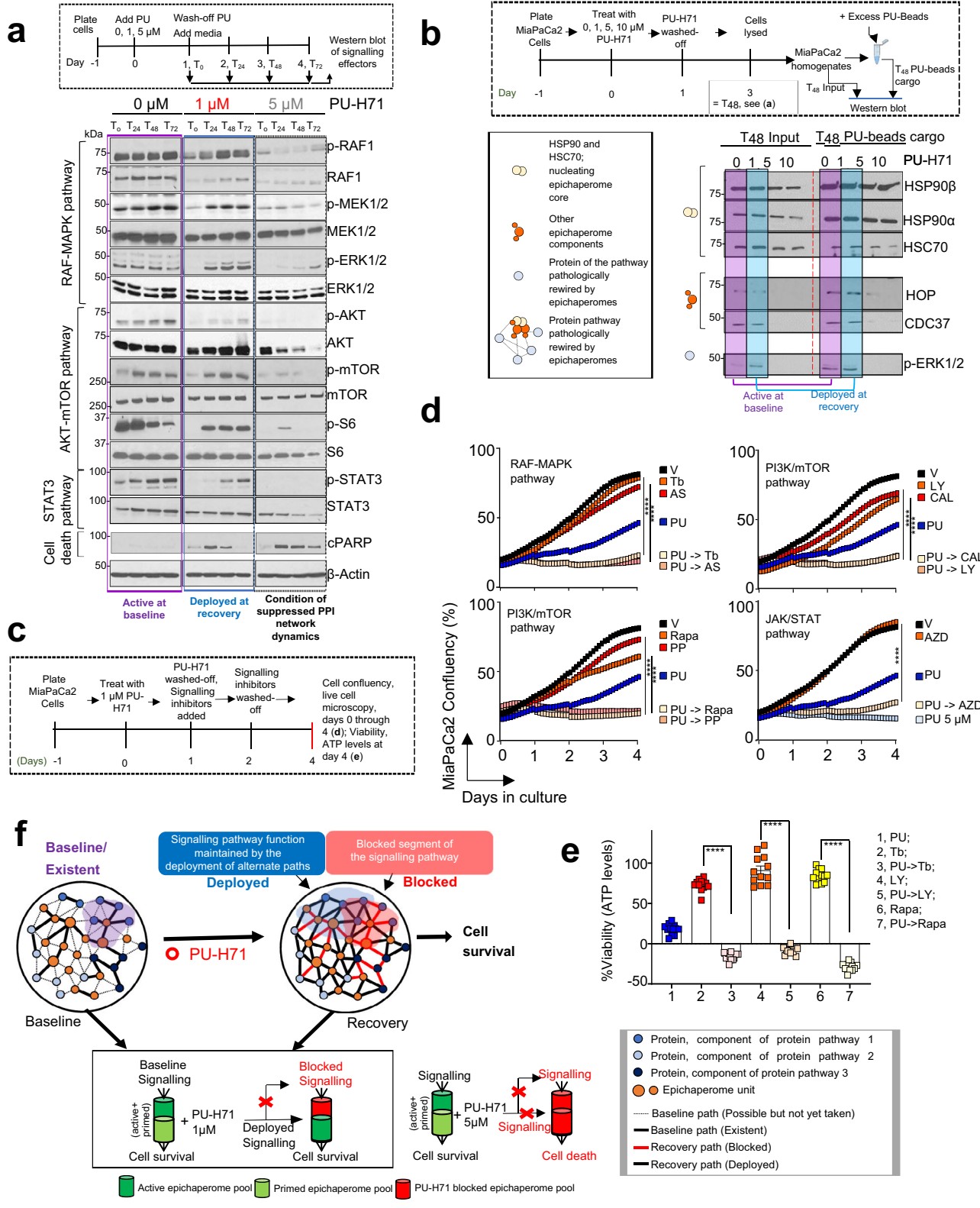

reduction in p-ERK levels, providing proof-of-principle validation that the pathway remains active at rebound and that its re-activation is executed at recovery through alternate paths (i.e., by engaging or co-opting also proteins other than those that may execute such functions at baseline).

To show that cellular rebound is executed through reactivation of protein pathways already existent, and constitutively active at baseline in MiaPaCa2 yet deployed at recovery through alternate

paths, we analysed the activity of signalling effectors (i.e., p-MEK, p-ERK, p-RAF1, p-AKT, p-S6, p-mTOR and p-STAT3) (Fig. 5a, b). We also tested the vulnerability of cells at recovery to inhibitors of these signalling pathways (see cell confluency and apoptosis in Fig. 5c–e). In MiaPaCa2 cells challenged with 1 μM PU-H71, these recovered from the $T_0$ levels ($T_0$, the time PU-H71 is removed from the culture) to baseline levels (i.e., levels seen in the 0 μM PU-H71 conditions at $T_0$ through $T_{72}$) in the period of

**Fig. 5 Epichaperome-mediated interactome hyperconnectivity retains baseline functionality of signalling networks. a** Western blot analyses of effectors of signalling pathways constitutively active in MiaPaCa2 (baseline) and at recovery (24 h to 72 h after 1 μM or 5 μM PU-H71 wash-off). **b** Western blot analysis of epichaperome components that establish interactome connections at baseline and at recovery (48 h after 1 μM PU-H71 ,wash-off). Treatment of cells with 5 and 10 μM PU-H71, where both active and primed epichaperome pools are blocked by PU-H71, and therefore no recovery is possible, i.e., no redundant paths are available (see also Fig. 1), is shown for comparison. **c** Schematic showing the experimental design. **d** Cell confluency monitored by live cell microscopy as in (**c**). Vehicle (V), DMSO; PU-H71 (PU), 1 μM; MEK1/2 inhibitors: Trametinib (Tb), 15 nM, AS703026 (AS), 1 μM; PI3K inhibitors: LY294002 (LY), 20 μM, CAL-101 (CAL), 10 nM; mTOR inhibitors: Rapamycin (Rapa), 30 nM, PP242 (PP), 80 nM; JAK inhibitor AZD1480 (AZD), 15 nM. Each curve represents the mean of three biological replicates treated with the indicated inhibitors, alone or in a sequential application, and is representative of three independent experiments; two-way ANOVA, Inhibitor vs PU-Inhibitor, ****$p < 0.0001$. PU-H71 5 μM, control for cell confluency in conditions of complete suppression of PPI network dynamics. **e** Same as in (**c**) for cell viability assessed by an ATP-based assay ($n = 12$ biological replicates from three individual experiments). Negative values represent a decrease in cell numbers compared to the initial cell population. Graph, mean ± SEM; unpaired two-tailed $t$ tests, Inhibitor vs PU- > Inhibitor, ****$p < 0.0001$. **f** Schematic summary of epichaperome-mediated signalling remodelling under pharmacologically-enforced interactome hyperconnectivity. Central to the present manuscript is which of the epichaperome pools is inhibitor bound (i.e. active? or primed? or both?) for either PPI network recovery or collapse. Herein, we manipulate these epichaperome pools through PU-H71 used in MiaPaCa2 cells at 1 μM (a concentration that engages mostly "active epichaperomes") and at 5 μM (a concentration that engages both "active epichaperomes" and "primed epichaperomes").

3 days (i.e., $T_{24}$ to $T_{72}$) monitored after removal of PU-H71 (Fig. 5a). Conversely, no signalling rebound was observed in cells challenged with 5 μM PU-H71 (Fig. 5a), the control condition of complete epichaperome suppression and thus of no rebound possible. Supportive of a retained dependence of these rebound signalling effectors on epichaperomes is their capture by the PU-beads at both baseline and recovery (see Fig. 5b, compare beads cargo at $T_{48}$ in cells released from either 0 or 1 μM PU-H71). Vulnerability of cells at recovery to inhibitors of baseline signalling pathways also supports the redeployment of same signalling pathways at recovery (Fig. 5c–e).

Thus, pharmacologically controlled, epichaperome-mediated, interactome hyperconnectivity reactivates protein pathways already in existence, and constitutively active, at baseline (Fig. 5f). Whereas new connections are established at the rebound, the goal of these PPIs is to retain overall constitutively active pathways, and in turn the baseline phenotype. Our findings support two important postulates. First, a 24 h exposure of cells to suboptimal PU-H71 concentrations creates a temporary state of interactome hyperconnectivity, and thus of maximal PPI network capacity where no further rebound is possible. Second, rebound though substantially rewiring PPI networks, is executed in such way that protein pathways intrinsically active remain active at recovery (Fig. 5f).

**Engineered hyperconnectivity provides vulnerability.** Therefore PU-H71-induced hyperconnectivity may sensitize tumours to inhibitors of such intrinsically active protein pathways, which has direct therapeutic implications (Fig. 6a). To test this potential outcome of our PPI controllability studies, we released cells treated with PU-H71 (1 μM for 24 h) into inhibitors of signalling pathways intrinsic to pancreatic cancer. We studied principally inhibitors tested in the context of pancreatic cancer in human patients or those already FDA approved[31,38]. We chose to study MiaPaCa2 and HPAF-II cell lines, both characterised at baseline by medium epichaperome levels, and thus of partly occupied PPI network capacity (ref. [18] and Fig. 1). We found inhibitors such as Pimasertib (AS-703026; MEK1/2 inhibitor), Trametinib (GSK1120212; MEK1/2 inhibitor), Apitolisib (GDC-0980, Class I PI3K and mTOR inhibitor), Torkinib (PP242, mTOR inhibitor), and LY29002 (broad-spectrum inhibitor of PI3K) had a modest effect in naïve MiaPaCa2 or HPAF-II pancreatic cancer cells (i.e., at baseline), as reported[31,38]. Importantly, each of these inhibitors became highly toxic to cells in the state of pharmacologically induced and epichaperome-mediated PPI hyperconnectivity (Fig. 6b). The reverse order of addition, i.e., MEK or PI3K inhibitor added before PU-H71 (Fig. 6c), and addition of the MEK

inhibitor at a stage hyperconnectivity has subsided (Supplementary Fig. 4), were both less toxic.

To understand how the results from the sequential treatment compare to the classical simultaneous combination treatment, we performed both live-microscopy monitoring to evaluate cell recovery, or lack of (Fig. 7a, b), as well as full isobologram treatment of PU-H71 (0–1 μM) and Trametinib (0–500 nM) to evaluate synergy (Fig. 7c). The simultaneous combination of PU-H71 and Trametinib was both cytotoxic and synergistic, which confirms the results of a recent study where it was identified as most active in a large-scale, unbiased in vivo screen of 57 different single agent and combination targeted therapies[44]. However, we observed cell recovery was substantially delayed in the sequential treatment paradigm when compared to the simultaneous drug-addition regimen. This was evidenced in the cell confluency curves recorded over the time cells were kept in culture following drug removal (Fig. 7a) and through images of the cell culture plate taken at day 7 (Fig. 7b). Supportive of distinct mechanisms of drug action, the synergy map over the dose matrix was also different, with the epichaperome-mediated PPI hyperconnectivity map showing high synergistic effect across most evaluated drug concentrations, whereas the simultaneous drug regimen was sensitive to the concentration of each agent in the combination (Fig. 7c).

We tested feasibility of the epichaperome-mediated PPI hyperconnectivity therapeutic approach in vivo and confirmed both target modulation and safety in the context of xenografted MiaPaCa2 tumours (MiaPaCa2 CDX) and patient-derived xenografts (PDX PC46), in mice (Fig. 8a). In tumours primed into interactome hyperconnectivity, Trametinib led to epichaperome collapse and suppression of tumour-supporting signalling activity (Fig. 8b, c, and Supplementary Fig. 5), which resulted in significant therapeutic effect of Trametinib (Fig. 8d) while retaining safety (Fig. 8e). Trametinib had little to no activity in naïve (i.e. baseline) tumours, as previously reported[31]. Poor efficacy of Trametinib and other MEK inhibitors in pancreatic cancer is caused by a reciprocal increase or maintenance in alternate signalling activity, such as the AKT/mTOR pathway activity[45]. We observed the deployment of such alternate paths was supressed when tumours were forced into the interactome hyperconnectivity state (Fig. 9a, b).

## Discussion
Our study provides proof-of-principle that through a mechanistic understanding of how PPI networks can be controlled through epichaperomes, we can manipulate drug resistance in the context of poor-outcome Ras-driven pancreatic cancers (Fig. 10). We

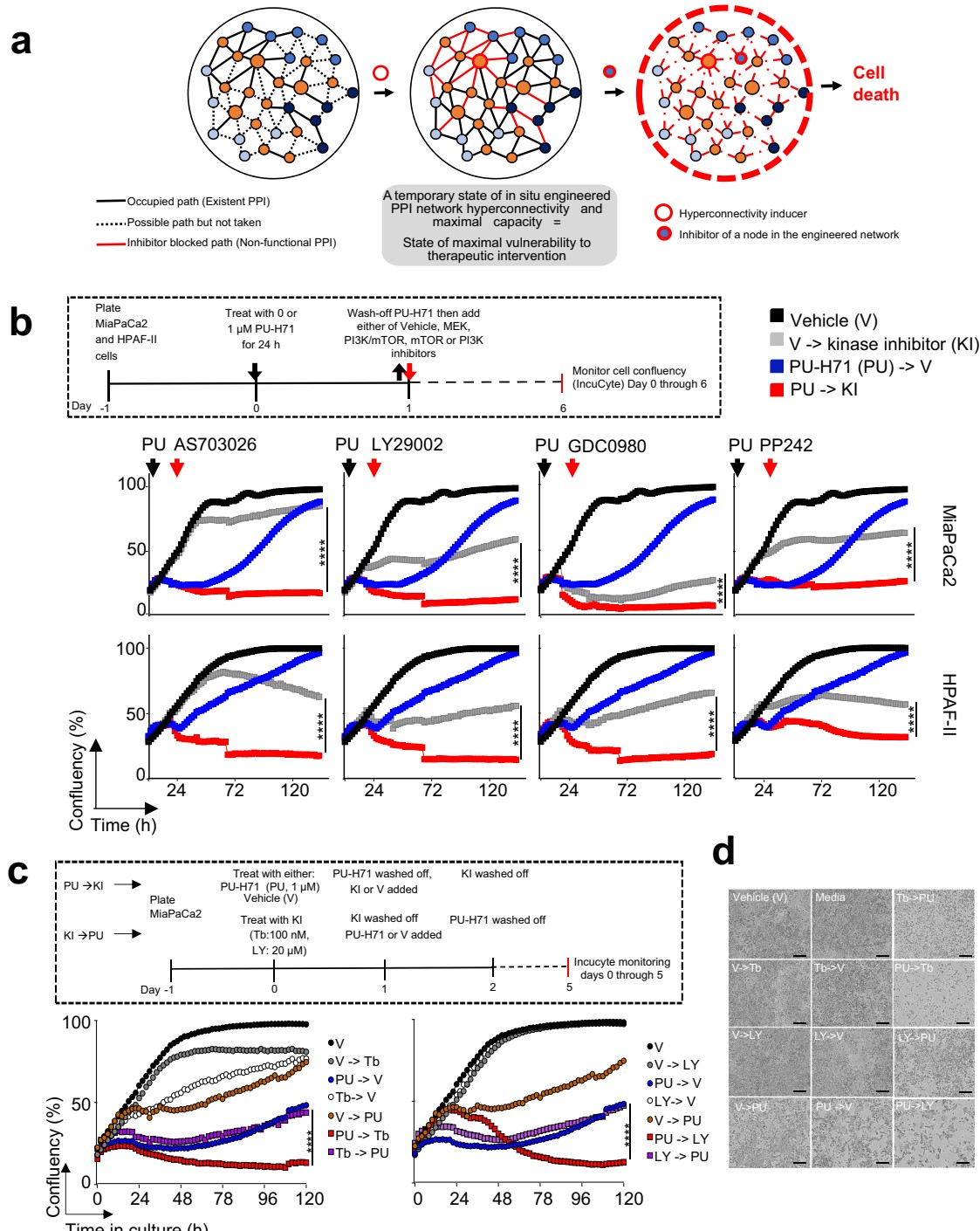

**Fig. 6 Current drugs are more effective than previously observed if controllability measures are exercised in tandem with drug treatment whereby cancerous cells are forced into a state of interactome hyperconnectivity. a** Schematic of the pharmacologically enforced, epichaperome-mediated interactome hyperconnectivity approach to create a state of maximal vulnerability for inhibitors of intrinsically active protein pathways. **b** Cell confluency monitored by live-cell microscopy under the treatment paradigm as in (**a**). MEK inhibitor AS703026, 2.5 μM; PI3K inhibitor LY29004, 10 μM; PI3K-mTOR inhibitor GDC0980, 2.5 μM; mTOR inhibitor PP242, 2.5 μM. Each curve represents the mean of 3 biological replicates treated with the indicated agents alone or in the sequential application; two-way ANOVA, V- > KI vs PU- > KI, ****$p < 0.0001$. **c, d** Pharmacologically-induced interactome hyperconnectivity imparts higher vulnerability than the sequential administration of kinase inhibitor followed by PU-H71. Cell confluency and viability monitored by live-cell microscopy. Each curve in (**c**) represents the mean of 3 biological replicates treated with the indicated agents alone or in the sequential application, two-way ANOVA, Inhibitor->PU vs PU- > Inhibitor, ****$p < 0.0001$. Micrographs in (**d**) are representative of each experimental condition from (**c**), day 5. V, vehicle; PU, PU-H71; Tb, trametinib; LY, LY29004. Scale bar, 300 μm.

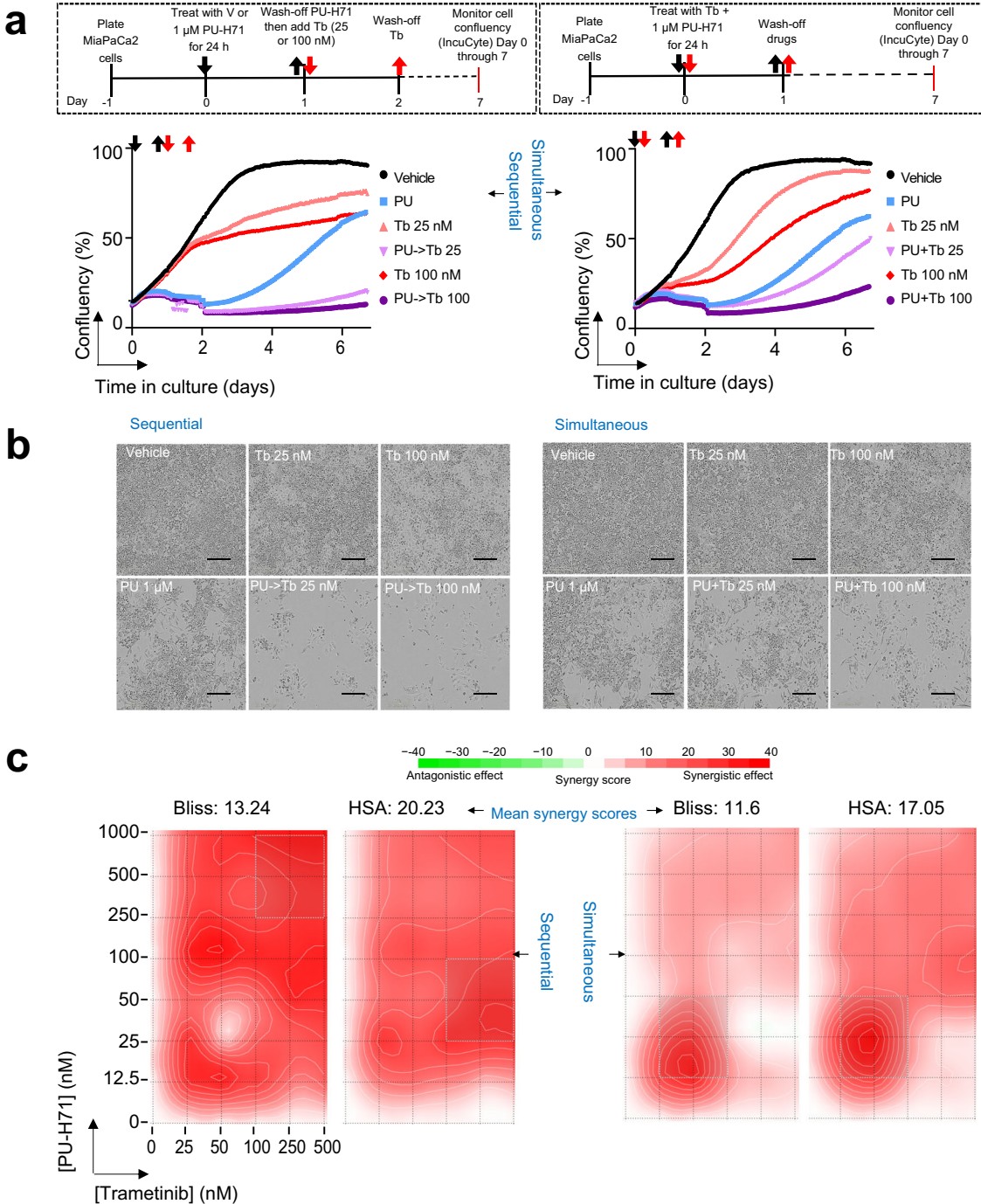

**Fig. 7 Comparison of sequential and simultaneous drug addition paradigms. a** Cell confluency monitored by live-cell microscopy under the treatment paradigm as shown in schematics. PU, PU-H71 at 1 μM; Tb 25 and Tb 100, trametinib at 25 nM and 100 nM, respectively. Each curve represents the mean of three biological replicates from three and four independent experiments for simultaneous and sequential treatment, respectively. **b** Micrographs are representative of each experimental condition from (**a**). Images were taken at day 7. Scale bar, 300 μm. **c** Synergy analysis for the drug paradigms shown in panel (**a**). ATP levels were measured at day 3 as a surrogate of cell viability. Synergy scores were calculated using the SynergyFinder web-application using two synergy scoring models. Bliss independence principle is appropriate when two drugs are mutually nonexclusive, i.e. when each targets a different pathway. The highest Single Agent (HSA) states that the expected combination effect equals to the higher effect of individual drugs.

propose controlling interactome connectivity through our approach has direct implications for broad-scale disease treatment by taking advantage of dynamic properties of cellular PPI networks to engineer a state of vulnerability in cells. This gestalt can be exploited to develop more effective and more specific therapies that target interactomes (as opposed to single genes or gene products). In this sense, our report serves as a steppingstone

for important future endeavours and creates a roadmap that can be utilized by independent research groups.

More specifically, we demonstrate the discovery of an approach for pharmacologically forcing the interactome into a state of hyperconnectivity and maximal PPI capacity for therapeutic vulnerability in cancer and present experimental evidence for interactome capacity engineering. This achievement, with proof-

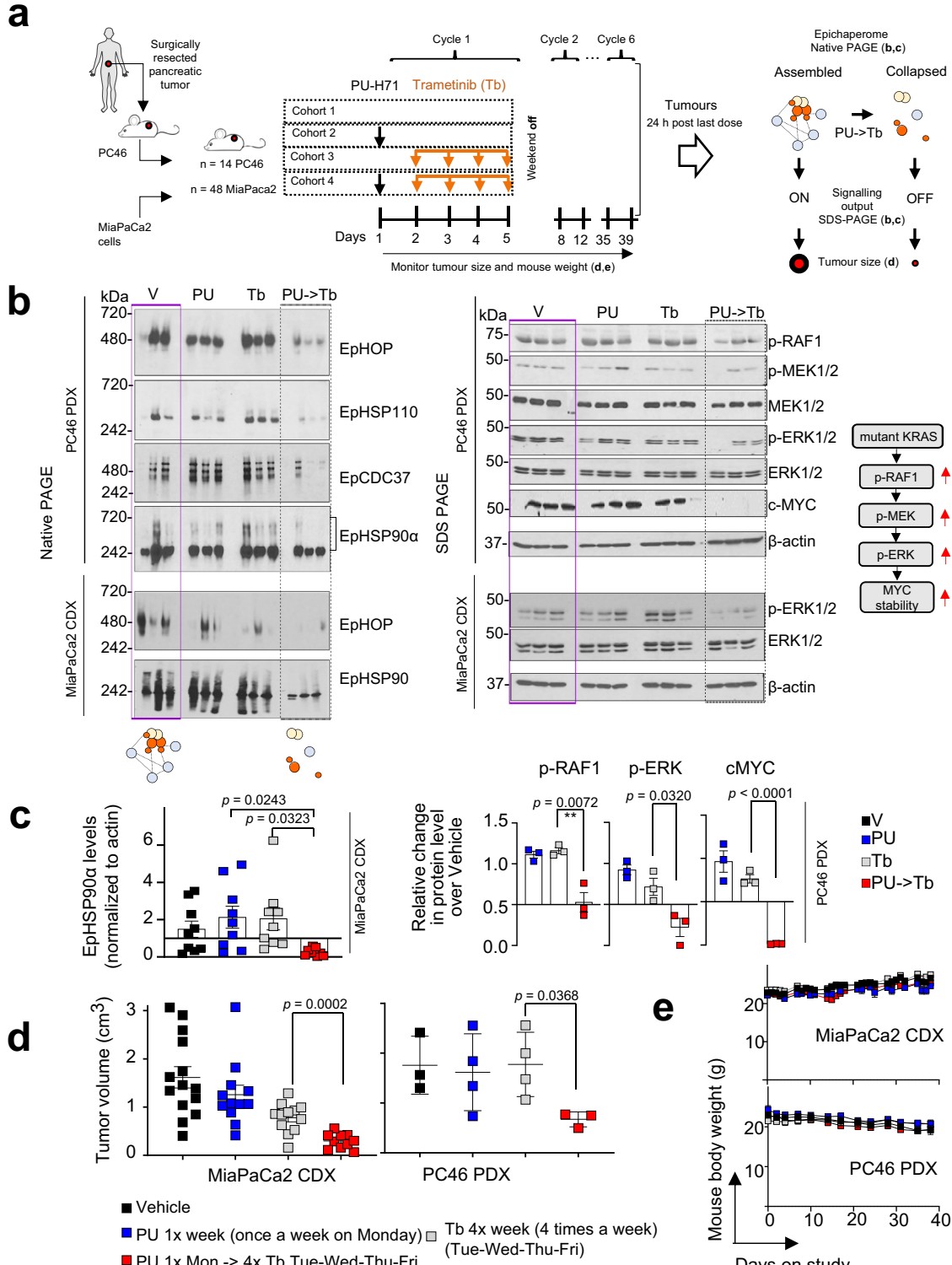

**Fig. 8 Trametinib becomes effective in Ras-driven pancreatic tumours when these are forced into a temporary state of interactome hyperconnectivity.**
**a** Schematic of the experimental design and the expected outcome of the PU-induced interactome hyperconnectivity. **b** Native-PAGE shows epichaperome levels and SDS-PAGE signalling activity in individual tumours ($n = 3$ from each cohort as in (**a**)). See also Supplementary Fig. 5. **c** Epichaperome levels in MiaPaCa2 tumours and signalling activity in PDX PC46 tumours. Error bars mean ± SEM; unpaired two-tailed $t$ test. **d** The anti-tumour activity of Trametinib (Tb) when tumours are forced into a state of interactome hyperconnectivity by PU-H71 (PU) given on Monday, each week. Tb is administered then daily, 24 h after PU. PU- > Tb, PU (75 mg kg$^{-1}$) given on the first day followed 24 h later by Tb (1 mg kg$^{-1}$) given on the next four consecutive days; Vehicle (Control); PU-H71 (75 mg kg$^{-1}$, PU) given once per week; Trametinib (1 mg kg$^{-1}$, Tb) given on four consecutive days. Pooled from 2 independent experiment, $n = 14$ and 3 mice for Vehicle; 12 and 4 mice for PU; 12 and 4 mice for Tb; 10 and 3 mice for PU- > Tb, in MiaPaCa2 CDX and PC46 PDX tumours, respectively. Error bars mean ± SEM; unpaired two-tailed t-test. **e** Body weight of each cohort as in (**a**). Graph, mean ± SEM.

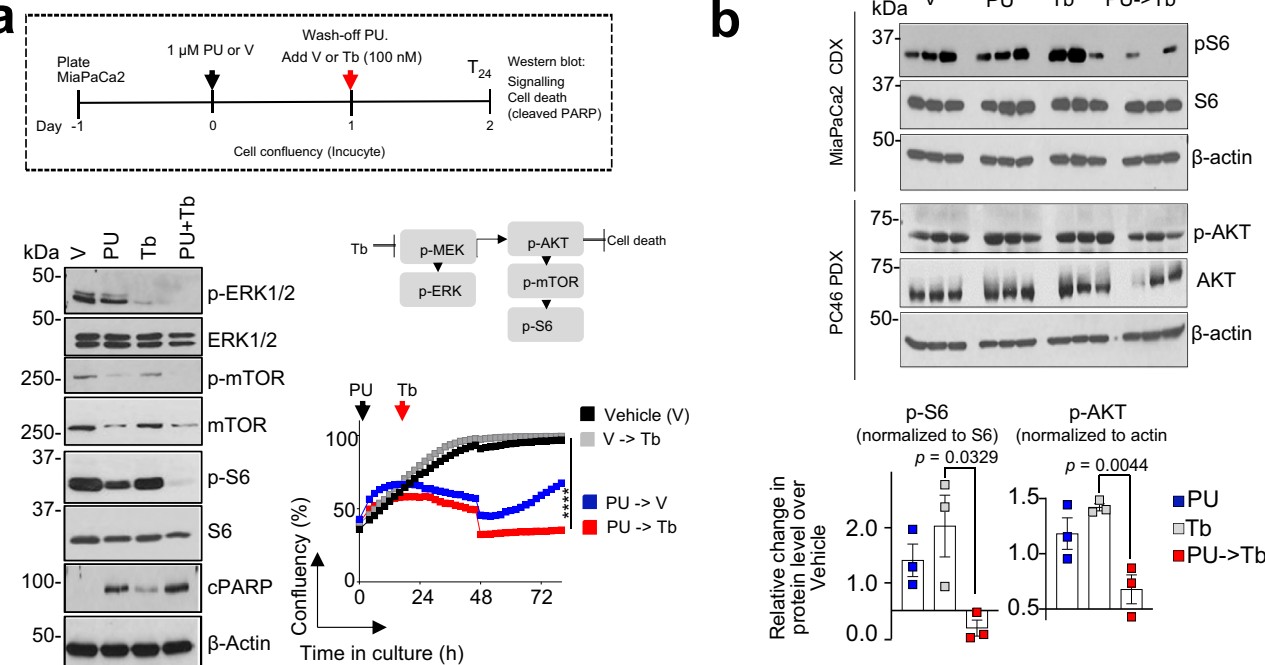

**Fig. 9 Feed-back mechanisms are suppressed in cells forced into a state of interactome hyperconnectivity. a** Schematic of the experimental design and Western blot analysis of signalling activity and cell death, as indicated. Representative data from three independent experiments are shown for Western blots. For cell confluency monitored by microscopy, each curve represents the mean of three biological replicates treated with the indicated agents alone or in the sequential application; two-way ANOVA, V- > Tb versus PU- > Tb, ****$p < 0.0001$. **b** Signalling activity in individual tumours ($n = 3$ from each cohort as in Fig. 8). Error bars mean ± SEM; unpaired two-tailed $t$ test.

of-principle provided here in pancreatic cancer, sets the stage for future efforts aiming to implement this strategy to other drugs and tumour types. Such knowledge may open the door for innovative, and feasible, cancer therapies based on understanding the capacity of interactome networks rather than trying to correct a pathway defect that has redundancy that lends itself to being refractory to drug treatments.

There is a fundamental difference between this approach—forcing interactomes into a state of maxed-out PPI capacity, i.e., dependent on a linear sequence of events, first, forcing into hyperconnectivity, second, treating at the hyperconnectivity stage—and synthetic lethality where simultaneous perturbation of two or more genes is required for cellular death. In synthetic lethality, two or more genes are present for redundancy. In hyperconnectivity, we pharmacologically engineer the proteomic state of refractory cancer cells to achieve therapeutic vulnerability. Therefore, our treatment paradigm is not a classical combination therapy method per se, because the hyperconnectivity inducer is used once to prime the tumour, followed by current therapy.

By providing here proof-of-principle using PU-H71 that interactome hyperconnectivity can be pharmaceutically modulated, we open the door for the search of interactome hyperconnectivity inducing agents. For example, future efforts could perform an unbiased screen for FDA-approved drugs to identify such hyperconnectivity-inducing agents, which will expand both the scope and clinical application of our findings.

Deciphering the complexity of changes within interactome networks in cancer is curbed by current technical limitations in the assessment of global changes in connectivity. This key limitation restricts our understanding of the disease. The present study takes advantage of our epichaperomics platform for systems level investigations of interactomes. This technique enables systems-level explorations of interactomes in endogenous (native) biological systems. By providing proof-of-principle in its use in

the study of interactome dynamics, we believe this approach will be a landmark method to engage PPI networks for therapeutic intervention by maxing out PPI network connectivity.

One aspect in controlling system dynamics is to predict where the deployment of new paths might occur. An important finding of this study is that deployment of new paths in our pharmacologically induced epichaperome-mediated hyperconnectivity approach retains the intrinsic activity of baseline protein networks. Bypass occurs through remodelling how intrinsic pathways are executed, rather than rerouting to alternate mechanisms. This suggests existing treatments may be more effective than previously observed if controllability measures are exercised in tandem with drug treatment whereby cells are forced into a state of hyperconnectivity, a finding of clinical implications. In the specific context of pancreatic cancer, a reassessment of small molecule inhibitors of downstream effectors of KRAS in the context of our chemically engineered vulnerability is therefore warranted based upon forcing interactome hyperconnectivity states. Another implication of the pharmacologically induced PPI hyperconnectivity approach is that a systemic analysis of the 'existent' and 'deployed' pathways intrinsic to individual tumours or subgroups of tumours could result in the creation of a knowledgebase of information for specific drug combinations that are likely more effective at treating particular cancer types.

We propose the in-situ engineering of PPI network connectivity is not limited to pancreatic cancer. An overarching treatment roadmap whereby interactome hyperconnectivity and subsequent vulnerability is engineered by pharmacologically modulating connectivity, altering cellular integration of protein networks, is a potential outcome of this work. It may open the door for innovative, and feasible, cancer therapies, where tumour priming with an interactome hyperconnectivity inducer renders tumours vulnerable to current therapies. By exploiting vulnerabilities in protein pathways for which many drugs are already available, treatment paradigms resulting from this study

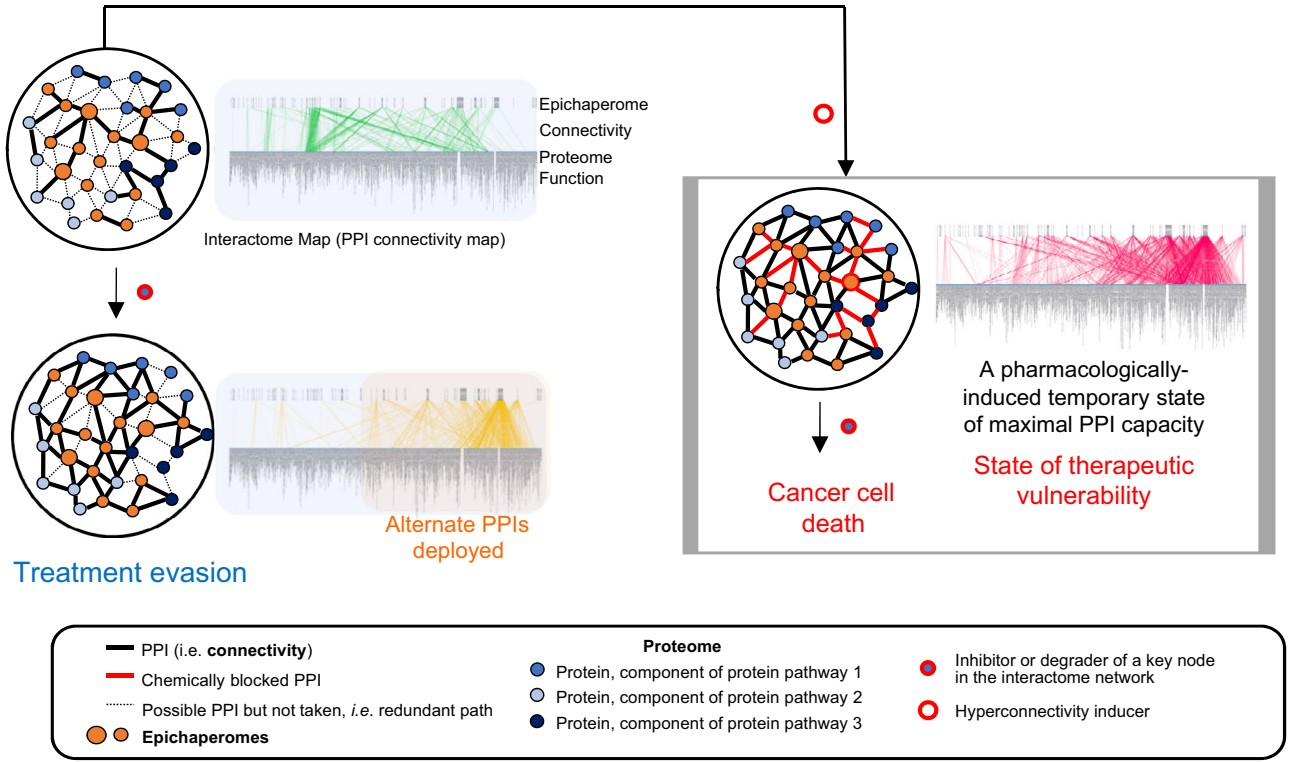

**Fig. 10 Summary of findings.** Interactome network plasticity in cancer, which arises from highly redundant signalling pathways, poses a real challenge to therapy and accounts for treatment resistance. Herein, we leverage discoveries in the biology of cellular stress linking cellular vulnerability to hyperconnectivity in PPI networks to propose a method for pharmacologically rewiring PPI networks at proteome-wide level for therapeutic vulnerability. By employing this approach, we provide proof-of-principle in cancer that by pharmacologically controlling interactome connectivity through epichaperomes, we anticipate the trajectory of PPI changes and co-opt dynamic cellular PPI networks to in situ engineer a state of hyperconnectivity. By forcing PPI networks into a hyperconnected state, we create therapeutic sensitivity, effectively making controllability into a powerful treatment strategy. This means existing treatments may be more effective than previously observed if controllability measures are exercised in tandem with drug treatment whereby cancerous cells are forced into a state of interactome hyperconnectivity. Several inhibitors/degraders of PPI nodes are already in clinical use or in development (ex. inhibitors or degraders such as PROTACS and others that target nodes in signalling networks), making this approach both timely and impactful. This treatment paradigm is not a combination method per se, because the hyperconnectivity inducer is used once to prime the tumour, followed by current therapy. It also differs from synthetic lethality where simultaneous perturbation of two or more genes is required for cellular death.

using existing drugs in concert with interactome controllability may have an immediate impact on the treatment and outcome of cancers. Approximately 50–60% of tumours express variable epichaperome levels, independent of tissue of origin, tumour subtype, or genetic background[18], confirmed presently in pancreatic tumour specimens, which are notorious for being treatment resistant. This suggests 50-60% of all tumours, independent of tumour type, could be engineered for interactome hyperconnectivity. Follow-through studies to provide evidence across various human cancer types and cancer subtypes, in those that express higher or lower epichaperomes, or tumours with mutations in genes or expressing higher levels of specific proteins that may affect the interactome priming will be needed to provide further mechanistic and therapeutic insights.

In the specific case of PU-H71 being used as the hyperconnectivity inducer, our data shows it is sufficient to employ PU-H71 once weekly (as the hyperconnectivity-inducer) followed by the ubiquitously used daily dose of a kinase inhibitor. This paradigm offers efficacy while retaining safety. Conversely, effective combinations with little toxic and off-target effects remain a challenge for traditional synthetic lethality-based combinations[46]. With PU-H71 in clinic, with activity and safety demonstrated in traditionally difficult to treat human cancers[25,26,33,47], our study offers treatment strategies of immediate clinical translatability.

In sum, we use innovative chemical biology approaches for systems-level analyses of proteome-wide protein connectivity to propose a method for engineering therapeutic vulnerability in cancer. We provide proof-of-principle in cancer by chemically controlling interactome connectivity, we co-opt dynamic interactome networks and in situ engineer a state of vulnerability. Leveraging this finding, we demonstrate how to enhance the effectiveness of currently available drugs. We show how existing treatments may be more effective than previously observed if controllability measures are exercised in tandem with drug treatments whereby cells are forced into a state of interactome hyperconnectivity. To the best of our knowledge, the approach of priming interactomes for leveraging vulnerability to improve treatment efficacy has not been done before.

## Methods

**Cell culture**. The human breast cancer, MDA-MB-468 (HTB-132; RRID:CVCL_0419) and pancreatic cancer cell lines; ASPC-1 (CRL-1682; RRID:CVCL_0512), PL45 (CRL-2558; RRID:CVCL_3567), MiaPaCa2 (CRL-1420; RRID:CVCL_0428), SU.86.86 (CRL-1837; RRID:CVCL_3881), CFPAC (CRL-1918; RRID:CVCL_1119), Capan-2 (HTB-80; RRID:CVCL_0026), BxPc-3 (CRL-1687; RRID:CVCL_0186), HPAF-II (CRL-1997; RRID:CVCL_0313), Capan-1 (HTB-79; RRID:CVCL_0237), Panc-1 (CRL-1469; RRID:CVCL_0480) and Panc 05.04 (CRL-2557; RRID:CVCL_1637) were purchased from the American Type Culture Collection (ATCC; Manassas, VA, USA). The patient-derived cells 931102 and 931019 were provided by Dr. Yelena Janjigian (MSKCC) and MSK-HR-Panc1 was provided by Dr. Vinagolu K. Rajasekhar (MSKCC). All cancer cell lines were obtained and cultured in Dulbecco's Modified Eagle's Medium (DMEM) supplemented with 10% foetal bovine serum (FBS), 100 U mL$^{-1}$ penicillin and 100 μg mL$^{-1}$ streptomycin at 37 °C in 95% humidified air with 5% $CO_2$. The human leukaemia cell line, HL-60 cell line (CCL-240; RRID: CVCL_0002), was purchased from ATCC and cultured in Iscove's Modified Dulbecco's supplemented with

20% FBS. All cell lines were tested for mycoplasma contamination and authenticated using short tandem repeat profiling prior to use in the study.

**Reagents**. PU-H71, PU-beads, control beads (CB), fluorescently labelled PU-H71 (PU-FITC), LSI-137 and the control derivative PU-FITC9 were synthesized using previously described protocols[18,48–50]. Signalling inhibitors used for in vitro experiments, such as, MEK/ERK pathway inhibitors, trametinib (GSK1120212) and pimasertib (AS-703026); ERK inhibitor, SCH779284; PI3K/mTOR pathway inhibitors, rapamycin, PP242 (Torkinib) and GDC0980 (Apitolisib); JAK/STAT inhibitor, AZD1480; AKT/PI3K inhibitors, LY294002, Idealisib (CAL-101, GS-1101); NFκB inhibitor, JSH-23; phospholipase-C inhibitor, U-73122 along with its inactive analogue, U-73343; and protein kinase A inhibitors; 6-22 amide and H-89, were purchased from Selleckchem. Trametinib used for animal studies was purchased from LC laboratories. Debio 0932 (CUDC-305) was purchased from ChemieTek.

**Human pancreatic tumour tissues**. All pancreatic tumour tissue specimens were obtained during clinically indicated collection procedures with informed consent and authorised through institutional review board (IRB)-approved bio-specimen protocol number 14-091 at Memorial Sloan Kettering Cancer Center. Samples were de-identified before receipt use in these approved studies. Samples were either used within 30 min to 1 h of collection for ex vivo studies or were minced, mixed with matrigel (50:50) and implanted subcutaneously in the flank of female NSG mice (Jackson Laboratory, Bar Harbour, NOD.Cg-Prkdcscid Il2rgtm1Wjl/SzJ, female, 20–25 g, 8 weeks old, IMSR Cat# JAX:005557, RRID: IMSR_JAX:005557) to generate PDX models, as previously described[51]. All mice were cared for in accordance with guidelines approved by the Memorial Sloan Kettering Cancer Center Institutional Animal Care and Use Committee under an approved IACUC protocol (04-03-009).

**Ex vivo studies in tumour explants**. Briefly, the fresh surgical samples were harvested in a sterile environment and delivered under 30 min from the surgical procedure. The sample was placed on wet ice and transported to the laboratory for ex vivo fresh tissue sectioning. Samples were then embedded in 4% Agarose gel and cut into 200 μm thick sections on a Leica VT 1000 S vibratome. The live sections were transferred into 24-well tissue culture plates and treated for 48 h with the indicated concentration of PU-H71 as previously described[52]. Following treatment, slices were fixed in 4% formalin solution for 1 h, then stored in 70% ethanol. For tissue analysis, slices were embedded in paraffin, sectioned, slide-mounted, and stained with haematoxylin and eosin (H&E). Apoptosis and necrosis of the tumour cells (as percentage) was assessed by reviewing all the H&E slides of the case (controls and treated ones) in toto, blindly, allowing for better estimation of the overall treatment effect to the tumour. Tissue slides were assessed by a pathologist who determined the apoptotic and necrotic events in the tumour, as well as any effect on adjacent normal tissue.

**In vivo studies in xenografted tumours in mice**. All animal studies were conducted in compliance with MSKCC's guidelines and under Institutional Animal Care and Use Committee (IACUC) approved protocols #05-11-024 and #04-03-009. Female athymic nu/nu mice (Hsd:Athymic Nude-Foxn1nu, female, 20-25 g, 6 weeks old; RRID: MGI:5652489) were obtained from Envigo and allowed to acclimatize at the MSKCC vivarium for 1 week before implanting tumours. Mice were provided with food and water ad libitum. All mice in all studies were observed for clinical signs at least once daily. Mice were housed in groups of 4-5 mice per individually ventilated cage in a 12 h light/dark cycle (6 am/6 pm), with controlled room temperature (22 ± 1 °C) and humidity (30-70%). Mice were provided with food and water ad libitum. All mice in all studies were observed for clinical signs at least once daily. Tumours were initiated by subcutaneous injection of $5 \times 10^6$ cells for MiaPaCa2 in a 200 μL cell suspension of a 1:1 v/v mixture of PBS with reconstituted basement membrane (BD Matrigel, Collaborative Biomedical Products). For PDX tumour implantation, tumour samples were cut into 3–4 mm pieces and immediately placed in DMEM medium supplemented with 10% foetal bovine serum and 100 units mL$^{-1}$ penicillin and 100 μg mL$^{-1}$ streptomycin. Tumour tissue pieces were embedded into Matrigel before subcutaneous implantation into mice. Once palpable, tumour volumes were calculated with calipers using the formula: Length × width$^2$ × 0.5. After tumours reached 0.1–0.2 cm$^3$ in size, animals were sorted and randomized for drug assignment, to achieve equal distribution of tumour size in all treatment groups. Before administration, a solution of PU-H71 and trametinib was formulated in citrate buffer and 10% NMP (10% 1-methyl-2-pyrrolidinone: 90% PEG-300), respectively. Animals were treated with vehicle alone, PU-H71 (75 mg kg$^{-1}$, i.p. on Monday), trametinib (1 mg kg$^{-1}$, oral gavage on Tuesday through Friday) or PU-H71 (75 mg kg$^{-1}$, i.p. on Monday) followed by trametinib (1 mg kg$^{-1}$, oral gavage on Tuesday through Friday). Body weights for all animals were recorded once per week during the administration of the test article. All mice were observed for clinical symptoms at the time the animals were received and on all days in which the test article was administered.

**Western blots**. Protein was extracted from cultured cells and xenograft tumours in 20 mM Tris pH 7.4, 150 mM NaCl, 1% NP-40 buffer with protease and

phosphatase inhibitors added (Complete tablets and PhosSTOP EASYpack, Roche) and centrifuged at 13,000 x g for 15 min at 4 °C. Protein concentrations were measured by using the BCA assay according to the manufacturer's protocol (Pierce™ BCA Protein Assay Kit, Thermofisher Scientific, Waltham, MA). Ten to fifty μg of total protein was resolved in acrylamide gels by SDS-PAGE, and then transferred to nitrocellulose membranes, blocked for 1 h in 5% milk in TBS, and incubated overnight with indicated validated primary antibodies. The membranes were blotted with HSP90α (ab2928; RRID:AB_303423; 1:6,000), HSP70 (ab94368; RRID:AB_10716913; 1:5,000) and XIAP (ab21278; RRID:AB_446157; 1:1,000) from Abcam; HSP90β (SMC-107; RRID:AB_854214; 1:2,000) and HSP110 (SPC-195; RRID:AB_2119373; 1:1,000) from Stressmarq; HSP70 (SPA-810; RRID:AB_10616513; 1:1,000), HSC70 (SPA-815; RRID:AB_10617277; 1:1,000), and HOP (SRA-1500; RRID:AB_10618972; 1:1,000) from Enzo; cleaved PARP (or PARP p85 Fragment pAb) (G7341; RRID:AB_430876; 1:1,000) from Promega; p-AKT (S473) (9271; RRID:AB_329825; 1:1,000), AKT (9272; RRID:AB_329827; 1:2,000), p-RAF1 (S259) (9421; RRID:AB_330759; 1:1,000), RAF1 (12552; RRID:AB_2728706; 1:500), p-MEK1/2 (S217/221) (9154; RRID:AB_2138017; 1:1,000), MEK1/2 (9122; RRID:AB_823567; 1:1,000), p-ERK1/2 (T202/Y204) (4377; RRID:AB_331775; 1:1,000), ERK1/2 (4695; RRID:AB_390779; 1:2,000), p-STAT3 (Y705) (9145; RRID:AB_2491009; 1:1,000), STAT3 (9139; RRID:AB_331757; 1:2,000), p-p65 (Ser536) (3033; RRID:AB_331284; 1:500), p65 (8242; RRID:AB_10859369; 1:2,000), p-mTOR (S2448) (5536; RRID:AB_10691552; 1:500), mTOR (2983; RRID:AB_2105622; 1:1,000), CDC37 (4793; RRID:AB_10695539; 1:1,000), CHIP (2080; RRID:AB_2198052; 1:2,000), HOP (5670; RRID:AB_10828378; 1:1,000), p-S6 ribosomal protein (Ser235/236) (4858; RRID:AB_916156; 1:2,000), S6 ribosomal protein (2217; RRID:AB_331355; 1:3,000), c-MYC (5605; RRID:AB_19039385605; 1:300) and GAPDH (2118; RRID:AB_561053; 1:4,000) from Cell Signaling Technology and β-actin (A1978, RRID:AB_476692; 1:6,000) from Sigma-Aldrich. The blots were washed with TBS/0.1% Tween 20 and incubated with appropriate HRP-conjugated secondary antibodies (Southern Biotech, Birmingham, AL, USA). The chemiluminescent signal was visualized with Enhanced Chemiluminescence System (GE Healthcare) following manufacturer's instructions and quantified using image Studio Lite Ver. 5.2 (LI-COR Biosciences).

**Epichaperome determination by native PAGE**. Cultured cells were lysed in 20 mM HEPES pH 7.5, 50 mM KCl, 5 mM MgCl$_2$, 0.01% NP40, 20 mM Na$_2$MoO$_4$ buffer, containing protease and phosphatase inhibitors. PDX and CDX tumour tissues were lysed by homogenisation in 20 mM Tris pH 7.4, 20 mM KCl, 5 mM MgCl$_2$, 0.01% NP40 buffer containing protease and phosphatase inhibitors. Protein concentrations were measured, and native gel electrophoresis was performed. Here 10-60 μg of protein extracts were loaded onto 5.5 % native gel and resolved at 4 °C. The gels were immunoblotted following a transfer in 0.1% SDS-containing transfer buffer for 1 h. The antibodies used were: HSP90β (SMC-107; RRID:AB_854214; 1:2,000) and HSP110 (SPC-195; RRID:AB_2119373; 1:1,000) from Stressmarq; HSC70 (SPA-815; RRID:AB_10617277; 1:500), HOP (SRA-1500; RRID:AB_10618972; 1:1,000) from Enzo; HSP90α (ab2928; RRID:AB_303423; 1:6,000) and HSP70 (ab94368; RRID:AB_10716913; 1:5,000) from Abcam; CDC37 (4793; RRID:AB_10695539; 1:1,000) and HOP (5670; RRID:AB_10828378; 1:500), from Cell Signaling Technology.

**Epichaperome determination using the PU-FITC flow cytometry assay**. The PU-FITC assay was performed as previously described[34,36]. Briefly, cells were incubated with 1 μM PU-FITC at 37 °C for 4 h. Then cells were washed twice with PBS and the adherent cells were detached using 0.25% Trypsin. Cells were then washed twice with FACS buffer (PBS/0.5% FBS) and finally resuspended in FACS buffer containing 1 μg mL$^{-1}$ DAPI (ThermoFisher Scientific, Cat. No. D1306). HL-60 cells (CCL-240; RRID: CVCL_0002) were used as internal control to calculate fold binding for all cell lines tested. The mean fluorescence intensity (MFI) of PU-FITC in viable cells (DAPI negative) was measured by flow cytometry on LSRFortessa (BD Biosciences). The FITC derivative FITC9 was used as a negative control[34]. The acquired data was saved into a flow cytometry standard file and analysed using Flow Jo software (FlowJo LLC).

**Fluorescence polarisation assay**. Fluorescence polarisation (FP) assays were carried out in black 96-well microplates (Greiner Microlon Fluotrac 200). A stock of 10 μM PU-FITC was prepared in Felts buffer (20 mM Hepes (K), pH 7.3, 50 mM KCl, 2 mM DTT, 5 mM MgCl$_2$, 20 mM Na$_2$MoO$_4$, and 0.01% NP40 with 0.1 mg ml$^{-1}$ BGG). 200 μg of protein lysates were equally divided into Eppendorf tubes and treated in triplicate with vehicle or indicated concentrations of PU-H71 at room temperature for 10 min (Fig. 1h). To evaluate if epichaperome binding by PU-H71 is retained after PU-H71 wash-off from the tissue culture, 200 μg of protein lysates, from cells treated for 24 h with PU-H71 then collected either immediately after media was added (T$_0$) or at 48 h after media was added (T$_{48}$), were evaluated in the FP assay (Supplementary Fig. 2). PU-FITC (10 nM) was added to each well in a final volume of 100 μl Felts buffer. To account for background signal, buffer and PU-FITC only controls were included in each assay. The FP values in mP were measured every 5–10 min or as indicated. The assay window was calculated as the difference between the FP value recorded for the bound fluorescent tracer and the FP value recorded for the free

fluorescent tracer (defined as mP − mPf). Measurements were performed on a Molecular Devices SpectraMax Paradigm instrument (Molecular Devices, Sunnyvale, CA), and data were imported into SoftMaxPro6 and analysed in GraphPad Prism 8.

**Clonogenic assays**. For clonogenic assays, cells were seeded in triplicate into 6-well plates and allowed to adhere overnight in regular growth media. Cells were then cultured in the presence of indicated doses of PU-H71 in complete media for 24 h. Next day, the drug was washed off and fresh complete media was added. After 72 h, the remaining cells were fixed with methanol (1%) and formaldehyde (1%), and then stained with 0.5% crystal violet. Relative growth was quantified by densitometry after extracting crystal violet from the stained cells using 10% of acetic acid. Experiments were performed at least three times.

**Viability assays**. Cells ($5 \times 10^3$) were seeded onto 96-well plates (solid black plate; Corning). Next day, cells were treated for 24 h with indicated doses of PU-H71. Cells were then washed twice with media and viability assessment was done after 72 h. For combination studies, drugs targeting different signalling pathways were added for another 24 h and then washed off. Vehicle control and drug treated cells were cultured in fresh media for 48 h and then viability assays were performed using CellTiter-Glo Luminescent Cell Viability Assay (Promega) which generates a luminescent signal proportional to the amount of ATP present in lysed cells. For annexin V staining, cells were labelled with Annexin V-PE after PU-H71 treatment, as previously reported[53]. SynergyFinder 2.0 (https://synergyfinder.fimm.fi) was used to calculate the degree of combination synergy or antagonism using reference models that assume no interaction between drugs[54]. SynergyFinder 2.0 generates synergy scores and plots, where values>10 indicate synergism; values between -10 and 10 indicate additive effects; and < -10 indicates antagonism. Measurements were performed on a Molecular Devices SpectraMax Paradigm instrument (Molecular Devices, Sunnyvale, CA), and data were imported into SoftMaxPro6 and analysed in GraphPad Prism 8.

**IncuCyte cell growth measurements**. The IncuCyte live-cell microscopy system (Essen BioScience) was used to evaluate cell confluency. Briefly, the day before the experiment, $10 \times 10^4$ cells were plated in 6-well plates and incubated overnight to allow cells to adhere. At the day of the experiment, the media were exchanged with complete media containing 1 μM PU-H71 or DMSO. After 24 h, cells were washed thrice with the PBS, fresh complete media was added, and cell growth was monitored for another 72 h. For combination studies, kinase inhibitors were added for 24 h following PU-H71 wash-off and cell growth was monitored for the indicated time intervals. Frames were captured from nine separate regions per well using a 20× objective. Confluence was measured using the IncuCyte software, where values were pooled, and the mean used to plot each datapoint on the graph. Data were imported and analysed in GraphPad Prism 8.

**Chemical precipitation**. For the chemical precipitation experiment, cells were lysed in the Felts buffer (20 mM Hepes pH 7.3, 50 mM KCl, 5 mM $MgCl_2$, 20 mM $Na_2MoO_4$, and 0.01% NP40 buffer) and quantified using the BCA assay according to the manufacturer's protocol (Pierce™ BCA Protein Assay Kit, Thermofisher Scientific, Waltham, MA), and 250–300 μg of total protein was incubated with either PU-beads or control beads (40 μL) for 3-4 h at 4 °C. Following centrifugation, beads and supernatant were separated, and beads washed thrice with the Felts buffer (20 mM Hepes pH 7.3, 50 mM KCl, 5 mM $MgCl_2$, 20 mM $Na_2MoO_4$, and 0.01% NP40 buffer). Bead cargo was eluted in Laemmli sample buffer to analyse the captured proteins using western blotting. For sequential chemical capture experiments, the supernatant was collected and subjected to three sequential rounds of chemical precipitation. The collected supernatants from each round were quantified using BCA assay. Proteins were loaded and resolved under both native and denaturing conditions and transferred as described above. Membranes were blotted with the HSP90α antibody (ab2928; RRID:AB_303423; 1:6,000) from Abcam, HSP90β (SMC-107; RRID:AB_854214; 1:2,000) and HSP110 (SPC-195; RRID:AB_2119373; 1:1,000) from Stressmarq; HSP70 (SPA-810; RRID:AB_10616513; 1:1,000) and HSC70 (SPA-815; RRID:AB_10617277; 1:1,000) from Enzo; p-ERK (T202/Y204) (4377; RRID:AB_331775; 1:1,000), CDC37 (4793; RRID:AB_10695539; 1:1,000) and HOP (5670; RRID:AB_10828378; 1:1,000), followed by anti-rabbit HRP-conjugated secondary antibody from Santa Cruz Biotechnology. Blots were visualized by enhanced chemiluminescence (GE). ImageJ (versions 1.4 and 1.52) was used for western blot quantification.

**Sample preparation for transcriptomics**. MiaPaCa-2 cells were treated with 1 μM of PU-H71 for 24 h, followed by drug washout and replacement with fresh medium. After 24 h, cell pellets were collected. RNA extraction, library preparations, sequencing reactions and bioinformatic analysis were conducted at GENEWIZ, LLC. (South Plainfield, NJ, USA) using the company's in-house protocols, as described below.

**RNA extraction, library preparations, sequencing reactions and bioinformatic analysis**. Total RNA was extracted using Qiagen RNeasy Plus Universal mini kit following manufacturer's instructions (Qiagen, Hilden, Germany). Extracted RNA samples were quantified using Qubit 2.0 Fluorometer (Life Technologies, Carlsbad, CA, USA) and RNA integrity was checked using Agilent TapeStation 4200 (Agilent Technologies, Palo Alto, CA, USA). RNA sequencing libraries were prepared using the NEBNext Ultra II RNA Library Prep Kit for Illumina following manufacturer's instructions (NEB, Ipswich, MA, USA). Briefly, mRNAs were first enriched with Oligo(dT) beads. Enriched mRNAs were fragmented for 15 min at 94 °C. First-strand and second-strand cDNAs were subsequently synthesized. cDNA fragments were end repaired and adenylated at 3'ends, and universal adapters were ligated to cDNA fragments, followed by index addition and library enrichment by limited-cycle PCR. The sequencing libraries were validated on the Agilent TapeStation (Agilent Technologies, Palo Alto, CA, USA), and quantified by using Qubit 2.0 Fluorometer (Invitrogen, Carlsbad, CA) as well as by quantitative PCR (KAPA Biosystems, Wilmington, MA, USA). The sequencing libraries were clustered on 1 flowcell lane. After clustering, the flowcell was loaded on the Illumina HiSeq instrument (4000 or equivalent) according to manufacturer's instructions. The samples were sequenced using a 2x150bp Paired End (PE) configuration. Image analysis and base calling were conducted by the HiSeq Control Software (HCS). Raw sequence data (.bcl files) generated from Illumina HiSeq was converted into fastq files and de-multiplexed using Illumina's bcl2fastq 2.17 software. One mismatch was allowed for index sequence identification. After investigating the quality of the raw data, sequence reads were trimmed to remove possible adapter sequences and nucleotides with poor quality using Trimmomatic v.0.36. The trimmed reads were mapped to the Homo sapiens reference genome available on ENSEMBL using the STAR aligner v.2.5.2b. The STAR aligner is a splice aligner that detects splice junctions and incorporates them to help align the entire read sequences. BAM files were generated as a result of this step. Unique gene hit counts were calculated by using feature Counts from the Subread package v.1.5.2. Only unique reads that fell within exon regions were counted. After extraction of gene hit counts, the gene hit counts table was used for downstream differential expression analysis. Using DESeq2, a comparison of gene expression between the groups of samples was performed. The Wald test was used to generate p-values and Log2 fold changes. Genes with adjusted $p$ values < 0.05 and absolute log2 fold changes > 1 were called as differentially expressed genes for each comparison. A gene ontology analysis was performed on the statistically significant set of genes by implementing the software GeneSCF.

**Differential gene expression analysis and gene set enrichment analysis on RNA-seq data**. The R package DESeq2 was used to determine the significant differentially expressed genes and calculate the log2 fold change of normalised mean hit counts in MiaPaCa-PU versus MiaPaCa. Gene annotations were downloaded from hsapiens_gene_ensembl database (version GRCh38.p13) via R package biomaRt (version 2.48.3) and all ensemble gene ids were mapped to hgnc symbol IDs for further analysis. For those hgnc symbols with multiple corresponding ensemble IDs, we aggregated them and took the mean value. Then, a universal gene set enrichment analysis was conducted via R package clusterProfiler (GSEA function; version 4.0.5), for which the DE gene set were pre-ranked by their log2 fold change values was tested against three annotated transcript factor gene sets downloaded from Enrichr libraries (https://maayanlab.cloud/Enrichr/; TRRUST_-Transcription_Factors_2019_enricher, ChEA_2016_enricher, ENCODE_TF_ChIP-seq_2015).

**Sample preparation for epichaperomics**. For PU-bait affinity purification, protein extracts were prepared in 20 mM Tris pH 7.4, 20 mM KCl, 5 mM $MgCl_2$, 0.01% NP40 buffer with protease and phosphatase inhibitors added (Roche). Samples were incubated with the PU-bait or the Control-bait for 3 h at 4 °C, washed with 20 mM Tris pH 7.4, 20 mM KCl, 5 mM $MgCl_2$, 0.01% NP40 buffer and subjected to SDS-PAGE. The samples were applied onto SDS-PAGE. The gels were stained with SimplyBlue Coomassie stain (ThermoFisher Scientific) and submitted for analysis. Gel lanes were cut into an average of 12 gel bands. Gel bands were completely destained with 50% methanol and 25 mM $NH_4HCO_3$ / 50% acetonitrile and diced into small pieces and dehydrated with acetonitrile and dried by vacuum centrifugation. The gel pieces were rehydrated with 10 ng μL$^{-1}$ trypsin solution (Trypsin Gold, Mass Spectrometry Grade, Promega) in 50 mM $NH_4HCO_3$ and incubated at 37 °C overnight. Peptides were extracted twice with 5% formic acid / 50% acetonitrile followed by final extraction with acetonitrile. Samples were concentrated to a very small volume by vacuum centrifugation and peptides reconstituted in 2% acetonitrile / 4% FA.

**Mass spectrometry data acquisition**. LC-MS/MS analysis was performed using a High Field Q Exactive mass spectrometer coupled to a Thermo Scientific EASY-nLC 1000 (Thermo Fisher Scientific, Waltham, MA) equipped with a self-packed 75 μm x 20-cm reverse phase column (ReproSil-Pur C18, 3 μm, Dr. Maisch GmbH, Germany) for peptide separation. Analytical column temperature was maintained at 50 °C by a column oven (Sonation GmBH, Germany). Peptides were eluted with a 3–40% acetonitrile gradient over 60 min at a flow rate of 250 nL min$^{-1}$. The mass spectrometer was operated in data-dependent (DDA) mode with survey scans acquired at a resolution of 120,000 over a scan range of 300-1750 m/z. Up to fifteen most abundant precursors from the survey scan were selected with an isolation window of 1.6 Th and fragmented by higher-energy collisional dissociation with

Normalised Collision Energies (NCE) of 27. Maximum ion injection time for the survey and MS/MS scans was 60 ms and the ion target value for both scan modes was set to 3e6.

**Data processing**. All mass spectra were first converted to mgf peak list format using Proteome Discoverer 1.4 and the resulting mgf files searched against a human UniProt protein database using Mascot (Matrix Science, London, UK; version 2.5.0; www.matrixscience.com). Decoy protein sequences with reversed sequence were added to the database to allow for the calculation of false discovery rates (FDR). The search parameters were as follows: (*i*) up to two missed tryptic cleavage sites were allowed; (*ii*) precursor ion mass tolerance = 10 ppm; (*iii*) fragment ion mass tolerance = 0.8 Da; and (*iv*) variable protein modifications were allowed for methionine oxidation, deamidation of asparagine and glutamines, cysteine acrylamide derivatization and protein N-terminal acetylation. MudPit scoring was typically applied using significance threshold score $p < 0.01$. Decoy database search was always activated and, in general, for merged LS-MS/MS analysis of a gel lane with $p < 0.01$, FDR averaged around 1%. Mascot search results were imported into Scaffold (Proteome Software, Inc., Portland, OR; version 4.7.3) to further analyse tandem mass spectrometry (MS/MS) based protein and peptide identifications. X! Tandem (The GPM, thegpm.org; version CYCLONE (2010.12.01.1) was then performed and its results were merged with those from Mascot. The two search engine results were combined and displayed at 1% FDR. Protein and peptide probability were set at 95% with a minimum peptide requirement of 1. In each of the Scaffold files that validate and import Mascot searched files, peptide matches, scoring information (Mascot, as well as X! Tandem search scores) for peptide and protein identifications, MS/MS spectra, protein views with sequence coverage and more, can be accessed. To read the Scaffold files, free viewer software can be found at: http://www.proteomesoftware.com/products/free-viewer. Mass spectra files were also analysed using the MaxQuant proteomics data analysis workflow (version 1.6.0.1) with the Andromeda search engine. Raw mass spectrometer files were used to extract peak lists which were searched with the Andromeda search engine against human or mouse proteome and a file containing contaminants, such as human keratins. Trypsin specificity with 2 missed cleavages with the minimum required peptide length was set to be seven amino acids. N-acetylation of protein N-termini, oxidation of methionines and deamidation of asparagine and glutamines were set as variable modifications. For the initial main search, parent peptide masses were allowed mass deviation of 20ppm. Peptide spectral matches and protein identifications were filtered using a target-decoy approach at a false discovery rate of 1%.

**Bioinformatics analyses**. LFQ intensity (from Maxquant) was used for protein quantitation. Quantile normalisations were performed within the replicates of the same sample. Missing values were filled by the minimal raw intensity across all the replicates of the same sample type. All resulting MS1 LFQ intensity signals were log10 transformed before being subjected to further analyses. Differential connectivity (DC) analysis was carried out on the pre-processed LFQ intensity data by performing Student's $t$ test (two-sided) analyses comparing MDA-MB-468 versus MiaPaCa2 untreated, or MiaPaCa2-PU versus MiaPaCa2. Fold change was calculated as a ratio of mean values from samples. The chaperome member list was created based on the reported 332 human chaperome dataset. To prepare the PPI database, we combined all entries from BioGrid (v.3.4.160) and IntAct (version 05.2018) to create a dataset that inventories all documented human-human (Homo sapiens) PPIs. As a quality control for selecting valid PPIs, interactions annotated as "genetic" (or psi-mi:MI:0208 in IntAct) or those from experiments of "Co- localisation", "Genetic interference", "Synthetic Rescue", "Synthetic Growth Defect" and "Synthetic Lethality" were removed. Unless otherwise specified, all PPIs were based on the combined database, prepared as described above. The interactome network was built in Cytoscape v3.61 by importing all the proteins identified from the epicharomics samples, using the aforementioned PPI database. Spring-electric algorithm was applied to generate the network layout. The node size and node colour represent the log10 transformed $p$ values and the fold change values (FC), respectively, from the DC analysis (See section Differential connectivity analysis). Nodes were coloured in grey if DC.$p > 0.1$. For figure clarity, the network edges and node labels were not shown. To investigate the functionality of the chaperome's interactome, we designed the interactome gene-set enrichment analysis (iGESA) method. It takes information from both PPI and Pathway enrichment analyses into analysis. The method applies GSEA within each local interactome of a given chaperome member (through PPI), which summarizes the most prominent biological functions extended from the chaperome member. By this method an additional layer of annotations can be attributed to the chaperome members from the Gene Ontology (GO) terms of their direct interactors. For each chaperome members detected by epicharomics (see also chaperome members section), iterations of the five following steps were performed: (i) extract the interactors of the given chaperome members; (ii) filter out non-DC interactors (DC.$p > 0.1$); (iii) define the "direction" within the remaining proteins, using the fold change (FC) value (>0 or <0) from the DC analysis; (iv) for each DC "direction", perform Fisher's exact test on Gene Ontology terms associated with the DC proteins (from the step 3) of this "direction"; (v) link each enriched GO terms with the identifier of the chaperome member and the DC"direction". The Fisher's exact test was performed using R package "ReactomePA", "clusterProfiler". The $p$

values from the Fisher's exact test were adjusted by the method of Benjamini Hochberg (FDR). For visualization, the links between chaperome members and enriched GO terms with adjusted $p$ value $\leq 0.001$ were imported in Cystoscape v3.61 using the columns "Chaperome_ID" (Protein identifier of Chaperome member) and "Description" (GO terms) as interactor A and B. Associated data ($p$ values, GeneRatio, BgRatio, $q$ value and adjusted p-values) were imported as edge attributes. The edges were coloured according to the experimental condition. To generate the figure, the hierarchical layout (yFiles) was applied with edge bends and unconnected GO terms removed. Heatmaps were generated using the R package "complexheatmap".

**Statistics and reproducibility**. Data analysis, statistical testing and visualization were conducted in R (version 3.5.1; R Foundation for Statistical Computing) or Prism (version 8 or 9). Statistical significance was determined using either Student's $t$ tests (2-group comparisons) or ANOVA (multiple comparisons). Pearson's coefficient was determined to measure the statistical relationship between variables. Means and standard errors were reported for all results unless otherwise specified. A $p$ value of less than 0.05 was statistically significant. Unless otherwise indicated, data are representative of at least three independent experiments. No statistical methods were used to predetermine sample sizes, but these are similar to those generally employed in the field.

**Reporting Summary**. Further information on research design is available in the Nature Research Reporting Summary linked to this article.

## Data availability

Epicharomics LC-MS data that support the findings of this study have been deposited in MassIVE with the MSV000085630 accession number. Epicharomics-derived interactomes are provided in Supplementary Data 1. Epicharomics related files (dependent libraries, chaperomics proteinGroups input, PPI networks, pathway databases and epicharomics datasets), final data outputs and visualizations (by Cytoscape) and other Cytoscape files are available in GitHub (github.com/chiosislab/Chaperomics_controllability_2020) and zenodo[55]. The results of pathway enrichment analyses were also included into the Cytoscape files[55]. RNAseq dataset have been deposited in GSA-Human with the HRA001454 accession number. Associated analyses are provided in Supplementary Data 2. The source data underlying the figures can be accessed in Supplementary Data 3. Uncropped and unedited images of the membranes used for immunodetection are available in Supplementary Fig. 6–18. Reagents are available under an MTA with Memorial Sloan Kettering; address requests to G.C.

## Code availability

The code for iGSEA is available in Github (github.com/chiosislab/Chaperomics_controllability_2020) and zenodo[55]. It is also explained and cited in the appropriate methods section and ref. [22].

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

## Acknowledgements

We thank the Antitumour Assessment Core and our colleagues in the Departments of Surgery and Medicine at Memorial Sloan Kettering for providing the biospecimens for research. This work is supported in part by the US National Institutes of Health (NIH) (R01 CA172546, R56 AG061869, R01 CA155226, P01 CA186866, P30 CA08748, R01 AG067598, R56 AG072599, R01 AG074004, R01 GM145739S10 RR027990, U54 OD020355-01 and P50 CA192937), the David M. Rubinstein Pancreatic Cancer Center, Hirshberg Foundation for Pancreatic Cancer, Mr. William H. Goodwin and Mrs. Alice Goodwin and the Commonwealth Foundation for Cancer Research and the Experimental Therapeutics Center of the Memorial Sloan Kettering Cancer Center; T.W. was supported by the Lymphoma Research Foundation; G.C. was supported in part by the Steven A. Greenberg charitable trust and the Solomon programme.

## Author contributions

G.C., S.J. and E.D.G. conceived the study. S.J. and E.D.G. designed and performed the biochemical and functional validation studies. S.J., E.D.G. and C.X. performed the explant studies, with scoring and analysis performed by A.C., L.T. and A.G. T.W. and W.W. designed and performed the computational analyses and the data visualization. T.T., S.S., S.B., S.J. and S.G. performed the in vivo studies. H.E-B and T.A.N performed the mass spectrometry sample preparation and protein identification. S.G., L.L.C., A.R., S.M., C.D., N.P., P.P., P.Y., J.A. and M.M. performed experiments. Y.J., E. dS., K.H.Y., M.L., J.H.H., W.R.J., V.K.R., P.J.A., S.D.I, O.G-H., T.T., C.S., L.S. and N.P. provided reagents. G.C., S.D.G. and S.J. wrote the paper.

## Competing interests

Memorial Sloan Kettering Cancer Centre holds the intellectual rights to PU-H71 (PCT/US06/03676, PCT/US2012/045861). Samus Therapeutics Inc, of which G.C. has partial ownership, and is a member of its board of directors, has licensed PU-H71. G.C., T.T., N.P. and E.D.G. are inventors on the licensed intellectual property. All other authors declare no competing interests.
