## [Transparent Peer Review File · Communications Biology]

Reviewers' comments:

Reviewer #1 (Remarks to the Author):

In their manuscript Joshi et al describe a method to sensitize cancer cells for therapeutics. Central to their findings is the epichaperome concept which embodies long-lived oligomeric structures containing several chaperones/ co-chaperones. These oligomeric structures provide a platform on which several signaling proteins improperly interact co-relating with a maladaptive state or disease. An earlier finding that cancer cells with high epichaperome content are vulnerable to therapeutics forms the basis of the current manuscript. Authors focus on pancreatic cancers and identify that otherwise recalcitrant cancers with low epichaperome can be sensitized by a pre-treatment with an epichaperome-selective drug, thus making them vulnerable to a subsequent treatment with therapeutics such as kinase inhibitors. Authors employ detailed biochemical, systems-level as well as in vivo xenograft models to confirm their findings giving the study a broad scope. This data-dense manuscript contributes to an interesting and perhaps important spin on the epichaperome concept that may lay foundation for a new method of sensitizing tumors, and hence will likely appeal to a larger audience.

While the manuscript has biochemical and in vivo data that support most of the findings, the study needs to be further strengthened to address the critical issues raised below (in no particular order): The authors have used only one epi-chaperome inhibitor to make all their claims. It is important to have a control in their experiments by using a chaperone inhibitor that does not have the properties of the epi-chaperome inhibitor to really prove that the author's interpretation of engineering connectomes. Any Hsp90 inhibitor could be used in at least one critical experiment - e.g. the sensitization by pre-treatment in cell lines. While all experiments need not be repeated with such a control, the novelty of this study, i.e. pre-treatment for sensitization, certainly needs a control of non-epichaperome inhibitor.

It is likely that the epi-chaperome inhibitor stays locked inside cells even after replacement of the medium containing the inhibitor. This is especially true as the cells do not grow for 48 hours after pre-treatment, so there may not be any dilution. If this is the case, it is likely that the central tenet of the study - namely sequential inhibition of two pathways - may in fact be misleading. The first inhibitor may continue to act when the second inhibitor is added. The authors should prove that their pre-treatment with PU does not lead to longer residence time of PU after its removal. Alternatively, the authors could test if double inhibition (simultaneous) is not as effective as sequential inhibition.

In line with above, it may be prudent to check how long does the engineered connectome last in absence of the inhibitor. The authors should test the simple prediction that addition of second inhibitor after losing the engineered state should have no effect on cell viability. The temporal correlation between epichaperome level and susceptibility can provide useful independent control of authors' interpretation of their data. The authors may want to use the info in Fig 5b to guide their experimental design (perhaps adding the second inhibitor 120h after removing PU?).

Fig 1i: How does one interpret the blots of tHOP and tHSC70 in 1uM treated MDA-cells after 48/ 72 hours? The cells have lost viability, are they not growing or do they have very little HSC70 or HOP? Given the other data in the figure it looks like these cells are almost dead, in which case what's the point of this comparison? Is there any protein synthesis going on in these cells?

Fig 2b: Why are the blue and green networks different? They are from same cells grown in similar conditions.

Fig 2d: To address the issue of HSP70 function in re-wiring/ by-passing the connectome, the authors have to inhibit HSP70 after withdrawing PU not before. Alternatively, the interpretation is that HSP70 inhibitors also re-wire the connectome and sensitize the cells for a subsequent treatment.

Fig 3a: It is not clear how PPIs are generated? Is this an experiment done by authors or is this a superimposition of chaperomics on known PPI in literature. This must be clarified as the interpretations are very different!

Page 7, line 26: The data can not be interpreted that the pathways at baseline are supported by epichaperome - they are just bound to the oligomeric structure.

Fig 3d: Is there anything special about proteins that are deployed/ pre-existent/ unchanged by pre-treatment? Any specific folds in proteins/ oligomeric structures/ ligand-dependence?

Fig 4a: Is this a blot of total proteins or PU-bound proteins? Almost all proteins look the same in T72 in 0 and 1 uM. A quantification of enough replicates along with the representation of MS data for these proteins will be useful.

Fig 4b: There is no difference in T48 cargo in 0/ 1 PU. It is not clear what the authors want to conclude from this.

In general the text is extremely difficult to read for a reader who has not followed all the previous work from the lab, and is not familiar with atypical terms such as maxed-out, hyperconnectivity etc. It is in authors' interest to explain their blots and interpretation in simpler terms/ more words. The first two figures are info-intense whereas the last few figures are much easier to follow. A redistribution of the data could be considered to simplify the paper's message. Similarly, the abstract does not convey the data but is written to mostly communicate interpretations of the authors. Specific molecular details replacing diffuse, unclear phrases, would be critical for revision of the abstract.

Reviewer #2 (Remarks to the Author):

Main Findings

Joshi et al. employ a chemical modulation approach of the cellular interactome to a hyperconnectivity state and show that this is associated with the increased response of pancreatic cancer cell lines to specific drugs including those that target the MAPK-pathways and PI3K-mTOR pathway. Furthermore, they show a proof-of-concept that this chemical modulation of the interactome achieved via epichaperome inhibition by the small molecule PU-H71 may hold great clinical prospects in treating pancreatic cancer and among other cancers. This is because priming interactomes as they showed may increase the vulnerability of cancer cell to drugs and thus improve drug efficacy. Overall, the methods and results of this paper are well done and straightforward to follow. I think that the current paper holds great biomedical significance, but at the same time, we can only realise such relevance upon further experiments and follow-through results as I have highlighted in the comments to the authors.

Comments to Authors

Page 7, line 2: The authors state that "Thus, analogous to baseline interactome hyperconnectivity, observed in MDA-MB-468 and other tumours, chemically induced hyperconnectivity is executed by enhancing the number of interactions and interaction partners between components of the two major chaperone networks, HSP90 and HSP70".

Is the relative contribution of HSP90 and HSP70 toward the shift to the hyperconnectivity state the same across different cell lines of the same cancer types, and/or cancer of different origins? If not (which I suspect), in a follow-up study, the authors could investigate this across various cancer types, other cell lines and xenograft of each of the cancer types and show the impact of one HSP having a more significant influence toward chemically induced hyperconnectivity against the other. i.e., one of

the HSP may be specific to particular tissues, or more effective than the other, which may also be depended on the tissue, which ultimately may affect the utility of this approach to treat tumours of specific origins.

Page 8, line 15: "We chose FOR study MiaPaCa2 and HPAF-II cell lines, both characterized at baseline by medium epichaperome levels, and thus of partly occupied network capacity".

It appears that the preposition "FOR" in the sentence is incorrect. The authors should consider replacing it with "TO".

Page 8, line 21: Importantly, each of these inhibitors became highly toxic to cells in the state of engineered hyperconnectivity (Fig. 5b). The reverse order of addition, i.e., MEK or PI3K inhibitor added before 23 PU-H71, was less toxic (Fig. 5c).

Did the authors investigate whether this effect is synergistic or additive by treating the cell with both PU-H71 and a MEK or PI3K inhibitor simultaneously? There is a possibility that a significant amount of PU-H71 may remain (even after washing) within the cell or bound to some cellular proteins. Maybe, alternatively (more straightforward), the authors could show that the washing step in their approach completely depletes the cancer cells of PU-H71. I should state that whether the effect is synergistic, or additive does not in any way invalidate the authors finding on the impact of PU-H71 treatment followed by a kinase inhibitor on xenografted tumours in mice.

In their previous publication (ref. 18) which the authors have cited several times in the current paper, they show that "under conditions of stress, such as malignant transformation fuelled by MYC, the chaperome becomes biochemically 'rewired' to form a network of stable, survival-facilitating, high-molecular-weight complexes." In the same publication, the authors show two types of cell lines, type-1 and type-2, associated with MYC expression and that PU-H71 binds strongly to HSP90 in the type-1 cell lines. Here, in the current publication, the authors should have carried out their experiments on both a type-1 cell line and a type-2 cell line as the type of cell line may be a factor that influences the clinical utility of their approach. Again, this is something that the authors can show in a follow-up study.

Page 8, line 29, the authors state that "Trametinib had little to no activity in naïve ('baseline') tumours, as previously reported²⁸. Poor efficacy of Trametinib and other MEK inhibitors in pancreatic cancer is caused by a reciprocal increase or maintenance in alternate signalling activity, such as the AKT/mTOR pathway activity. We observed the deployment of such alternate paths was suppressed when tumours were forced into the interactome hyperconnectivity state."

This is the main punchline of the study as many studies show that a significant number of tumours are resistant to drugs because of alternative pathways being activated that lead to the activation of the same downstream transcription factors and mechanisms that the drug should abate. The "alternative signalling activity" and how the authors' approach takes care of this conundrum in cancer cells should even be mentioned in the abstract, if possible, as there is space for a few more words in their abstract.

About the title "Interactome engineering for therapeutic vulnerability in cancer".

I believe the title and to a lesser extent the abstract of the study undersell the importance of the authors' findings. I don't have a better title for the manuscript, but the journal Editor and the authors could develop a title that will give more prominence to the paper. This title should have some information on pharmaceutical modulation (not engineering) of cellular networks to a hyperconnectivity state leading to increased drug sensitivity. Furthermore, I think that "interactome engineering" part of the title may be somehow misleading (a riddle or brainteaser) that might undersell this paper's importance. Possibly a few simple and direct words would increase the visibility and readership of this great work.

The discussion section also starts with a "brainteaser" sentence: "We demonstrate the discovery of an approach for engineering the interactome into a state of hyperconnectivity and maxed-out capacity for

therapeutic vulnerability in cancer and present experimental evidence for interactome capacity engineering." I wish that the authors could write these sentences using simple and straight forward words like the two sentences that follow the opening statement of the discussion: "This achievement, with proof-of-principle provided here in pancreatic cancer, sets the stage for future efforts aiming to implement this strategy to other drugs and tumour types. Such knowledge may open the door for innovative, and feasible, cancer therapies based on understanding the capacity of interactome networks rather than trying to correct a pathway defect that has redundancy that lends itself to being refractory to drug treatments."

I agree with the authors' statement that "we believe this approach will be a landmark method to engage PPI networks for therapeutic intervention by maxing out hyperconnectivity". However, there should be follow-through studies to provide evidence across various human cancer types and cancer subtypes, those that express higher or lower MYC levels (i.e., type-1 and type-2), or tumours that have mutations in genes or express higher levels of specific proteins that may affect the interactome priming. The authors could also investigate how the primed cells respond to drugs that target other pathways (other than the MAPK and PI3K pathways) for which there is likely to be variations in drug efficacy.

The title says, "in cancer", but the authors only show a proof-of-concept for pancreatic cancer. Maybe the word cancer could be return for the sake of having a broader readership for the paper.

The protein-protein interaction network changes observed after chemical perturbation are likely linked to changes in transcription factor expression and activity. Can the author provide information on transcription factor enrichments (the Ma'ayan Lab's Enrichr tool can do this) between the baseline and hyperconnectivity state of the MiaPaCa cell line? Also, they could analyse for the enriched transcription factors between the baseline MiaPaCa cell line and the MDA-MB-468 cell line (which is already in a hyperconnectivity state) and see if the same transcription factors show up in the comparisons of the baseline MiaPaCa vs primed MiaPaCa and the baseline MiaPaCa vs MDA-MB-468.

The results shown in Figure 3d are quite remarkable! The authors show that new signalling routes are activated within several signalling pathways, including the MAPK pathway, PI3K pathway, TGF- β pathway and among others. It would be nice if the authors could also evaluate the effect of treating the cancer cell in the hyperconnectivity state with 1) a drug that targets a protein (e.g., a kinase) within the 'existent' signalling route, 2) a drug that targets a protein in the 'deployed' signalling route, and 3) a combination of both drugs. Furthermore, it would be nice if the authors could evaluate the effect of treating the unprimed (baseline) MiaPaCa cell line with a combination of drugs that target proteins within both the 'existent' and 'deployed' signalling route of a signalling pathway of their choosing. If this approach works similar to the priming followed by treatment approach, then the priming approach could be used to create a knowledgebase of information on 'existent' and 'deployed' in cancer and thus specific drug combinations that are likely more effective at treating particular cancer types.

Tissue or cell-type-specific variations in the wiring of cellular networks are attributable to the differences in the expression profile of proteins – specific proteins are not expressed in some tissues, mostly because of epigenetics. Therefore, it is improbable that the cell lines can shift from the baseline state to the hyperconnectivity state without changes in the epigenome, e.g., the proteins that were not previously expressed (in the baseline state) but required to link two or more proteins in a network must be present (in the hyperconnectivity state). As such, I suggest that the authors explore changes that occur in the epigenome in cancer cells between the baseline and the hyperconnectivity state. Such information would give us more valuable insights on how to treat cancers.

We thank the Reviewers for their thoughtful and constructive critiques, and are pleased they concluded this manuscript merits publication in *Communications Biology* pending revision. We are also grateful the Reviewers appreciate this “data-dense manuscript” that this submission “contributes to an interesting and perhaps important spin on the epichaperome concept that may lay foundation for a new method of sensitizing tumors, and hence will likely appeal to a larger audience”, and “holds great biomedical significance”.

Below we address in point-by-point fashion the comments and concerns of the Reviewers.

Reviewer #1: In their manuscript Joshi et al describe a method to sensitize cancer cells for therapeutics. Central to their findings is the epichaperome concept which embodies long-lived oligomeric structures containing several chaperones/ co-chaperones. These oligomeric structures provide a platform on which several signaling proteins improperly interact co-relating with a maladaptive state or disease. An earlier finding that cancer cells with high epichaperome content are vulnerable to therapeutics forms the basis of the current manuscript. Authors focus on pancreatic cancers and identify that otherwise recalcitrant cancers with low epichaperome can be sensitized by a pre-treatment with an epichaperome-selective drug, thus making them vulnerable to a subsequent treatment with therapeutics such as kinase inhibitors. Authors employ detailed biochemical, systems-level as well as in vivo xenograft models to confirm their findings giving the study a broad scope. This data-dense manuscript contributes to an interesting and perhaps important spin on the epichaperome concept that may lay foundation for a new method of sensitizing tumors, and hence will likely appeal to a larger audience.

While the manuscript has biochemical and in vivo data that support most of the findings, the study needs to be further strengthened to address the critical issues raised below (in no particular order):

1. The authors have used only one epi-chaperome inhibitor to make all their claims. It is important to have a control in their experiments by using a chaperone inhibitor that does not have the properties of the epi-chaperome inhibitor to really prove that the author’s interpretation of engineering connectomes. Any Hsp90 inhibitor could be used in at least one critical experiment - e.g. the sensitization by pre-treatment in cell lines. While all experiments need not be repeated with such a control, the novelty of this study, i.e. pre-treatment for sensitization, certainly needs a control of non-epichaperome inhibitor.

R1.1. Response: Thank you for this insightful comment. Accordingly, we now add additional data on the inhibitor Debio-0932 (CUDC-305) (Figure 2e,f and pasted below). We specifically chose this inhibitor because, albeit structurally similar to PU-H71, it only partially suppresses epichaperomes even when used at concentrations reported to result in maximal biological activity. We note most inhibitors that have moved to clinic in cancer and act by inserting into the ATPase pocket of HSP90 do in fact have some measurable effect on epichaperomes. What differentiates these inhibitors is, *i*) How well they suppress epichaperomes (please note the partial suppression shown in the revised submission for CUDC-305 and total suppression for PU-H71) and *ii*) How selective these inhibitors are for epichaperomes relative to HSP90 (Joshi et al. Nature Reviews Cancer 2018). The first feature determines drug efficacy whereas the latter influences its therapeutic index.

With these being said, the best control here is PU-H71 itself used in MiaPaCa2 cells at 1 μM (a concentration that engages mostly “active epichaperomes”) and at 5 μM (a concentration that engages both “active epichaperomes” and “primed epichaperome”), as we have used in the manuscript. We apologize this was not expressed more clearly in the initial submission. Accordingly, we reiterate the nature of these distinct

epichaperome pools and their relationship to cell recovery, or the lack thereof, as also pointed out by Reviewer #1 below. Although the original Figure 1 contained schematics explaining the nature of epichaperome pools, their binding preference for PU-H71 and their influence of network capacity, we could have done a better job explaining these results and their implications. We now *i)* provide further clarification in the revised text (Results, p. 5, blue lettering), and *ii)* include explanatory diagrams in Figure 1 (see Figures 1d-f). We also divided Figure 1 into two figures (now Figures 1 and 2) to provide greater clarity on these key points (page 5).

2. It is likely that the epichaperome inhibitor stays locked inside cells even after replacement of the medium containing the inhibitor. This is especially true as the cells do not grow for 48 hours after pre-treatment, so there may not be any dilution. If this is the case, it is likely that the central tenet of the study - namely sequential inhibition of two pathways - may in fact be misleading. The first inhibitor may continue to act when the second inhibitor is added. The authors should prove that their pre-treatment with PU does not lead to longer residence time of PU after its removal. Alternatively, the authors could test if double inhibition (simultaneous) is not as effective as sequential inhibition.

R1.2. Response: Thank you for suggesting this experiment and for encouraging us to clarify the text associated with Figure 1. We believe, as noted in R1.1, we did not fully or clearly explain the existence and significance of the active and primed epichaperome pools in relation to network hyperconnectivity. Indeed, PU-H71 becomes trapped when bound to epichaperomes (see trapping, Figure 1c and see Time zero, T_0 in Figures 2a), and thus stays locked, a step followed eventually by epichaperome disassembly (see Figure 1c, Time 24 and beyond for 1 and 5 μM PU-H71 in MDA-MB-468 and 5 μM PU-H71 in MiaPaCa2). We reported on this drug-target mechanism both pre-clinically and clinically (Rodina *et al.* Nature 2016, Pillarsetty *et al.* Cancer Cell 2019, Inda *et al.* Nature Communications 2020, Bolaender *et al.* Nature Communications 2021, Jhaveri *et al.* JCO Precision Oncology 2020).

Central to the present manuscript is which of the epichaperome pools is inhibitor bound (i.e., Figure 1; active? or primed? or both?) for either network recovery or collapse. Herein, we manipulate these epichaperome pools through PU-H71 used in MiaPaCa2 cells at 1 μM (a concentration that engages mostly “active epichaperomes”) and at 5 μM (a concentration that engages both “active epichaperomes” and “primed epichaperomes”), to derive our conclusions as we have state in the revised manuscript.

To this end, we added the following paragraph (p. 5, lines 3-15), “PU-H71 is an HSP90 inhibitor that kinetically selects for the epichaperome over HSP90 pools, as it becomes trapped when bound to HSP90 in epichaperomes whereas it exits rapidly from HSP90^{18,31}. The biochemical and functional mechanism of epichaperome modulation by PU-H71 comprises a first step of trapping of epichaperomes along with their interacting proteins^{18,22,33}. This step is followed by epichaperome collapse and restoration of normal PPI network connectivity^{20,22}, detrimental to both the maintenance of a malignant phenotype and cancer cell survival (Fig. 1c and ref. 18,19). Importantly, the dissociation rate constant (K_{off}) of PU-H71 from epichaperomes is proportional to epichaperome occupancy with a K_{off} from active epichaperomes (i.e. proteome bound) $<$ K_{off} from primed epichaperomes (i.e. not proteome bound) (Fig. 1d-f schematics and ref. 18). Proteome in this context is the full complement of proteins being bound to and impacted by epichaperomes in a specific cellular context. This indicates PU-H71 when used at well-defined concentrations, it can be used to preferentially engage and interrogate only active epichaperomes or both active and primed epichaperomes.” We also expanded the text associated with Figure 1 to provide further explanation on the incorporated data (page 5, blue lettering, lines 23-27).

With regard to ‘double inhibition (simultaneous) is not as effective as sequential inhibition’, which is also a comment made by Reviewer #2 (comment 2.3), we added this comparison to the newly created Figure 7 (see pasted on the next page).

The following text was added to accompany this figure, “To understand how the results from the sequential treatment compare to the classical simultaneous combination treatment, we performed both live-microscopy monitoring to evaluate cell recovery, or lack of (Fig. 7a,b), as well as full isobologram treatment of PU-H71 (0 to 1 μM) and Trametinib (0 to 500 nM) to evaluate synergy (Fig. 7c). The simultaneous combination of PU-H71 and Trametinib was both cytotoxic and synergistic, which confirms the results of a recent study where it was identified as most active in a large-scale, unbiased in vivo screen of 57 different single agent and combination targeted therapies⁴⁴. However, we observed cell recovery was substantially delayed in the sequential treatment paradigm when compared to the simultaneous drug-addition regimen. This was evidenced in the cell confluency curves recorded over the time cells were kept in culture following drug removal (Fig. 7a) and through images of the cell culture plate taken at day 7 (Fig. 7b). Supportive of distinct mechanisms of drug action, the synergy map over the dose matrix was also different, with the epichaperome-mediated PPI hyperconnectivity map showing high synergistic effect across most evaluated drug concentrations, whereas the simultaneous drug regimen was sensitive to the concentration of each agent in the combination (Fig. 7c).” Page 11, lines 5-19.

3. In line with above, it may be prudent to check how long does the engineered connectome last in absence of the inhibitor. The authors should test the simple prediction that addition of second inhibitor after losing the engineered state should have no effect on cell viability. The temporal correlation between epichaperome level and susceptibility can provide useful independent control of authors’ interpretation of their data. The authors may want to use the info in Fig 5b to guide their experimental design (perhaps adding the second inhibitor 120h after removing PU?).

R1.3. Response: This is an experiment with many caveats associated with keeping cells in a culture plate for a period of 2 weeks during live cell microscopy monitoring (eg. media evaporation, challenge of cells being disturbed at each media replacement step, and other). Nevertheless, we attempted this study and added the data to the newly created Supplementary Fig. 4. We included the following sentence in the text: “The reverse order of addition, *i.e.*, MEK or PI3K inhibitor added before PU-H71 (Fig. 6c), and addition of the MEK inhibitor at a stage hyperconnectivity has subsided (Supplementary Fig. 4), were both less toxic.”

4. Fig 1i: How does one interpret the blots of tHOP and tHSC70 in 1uM treated MDA-cells after 48/ 72 hours? The cells have lost viability, are they not growing or do they have very little HSC70 or HOP? Given the other data in the figure it looks like these cells are almost dead, in which case what's the point of this comparison? Is there any protein synthesis going on in these cells?

R1.4. Response: MDA-MB-468 was included as control for a cellular system comprised by majorly active epichaperome pools, revealing maxed-out PPI network capacity characterized by hyperconnected networks. It is included to show side-by-side the effect on cellular networks and cell viability in conditions of both total and partial epichaperome suppression. We added the above paragraph to the text (top, page 6, lines 3-7).

5. Fig 2b: Why are the blue and green networks different? They are from same cells grown in similar conditions.

R1.5. Response: The comparisons are performed pair-wise between MDA-MB-468 and MiaPaCa2 and between MiaPaCa2-PU and MiaPaCa2 (Fig. 2b). Each interactome map shows the differences between the individual pairs. For example, the map 'MiaPaCa vs MDA468' shows what is different between these two conditions whereas 'MiaPaCa-PU vs MiaPaCa' shows what is different between baseline and rebound MiaPaCa2. Therefore, we did not expect maps across comparisons to be identical.

6. Fig 2d: To address the issue of HSP70 function in re-wiring/ by-passing the connectome, the authors have to inhibit HSP70 after withdrawing PU not before. Alternatively, the interpretation is that HSP70 inhibitors also re-wire the connectome and sensitize the cells for a subsequent treatment.

R1.6. Response: We respectfully disagree with this interpretation. LSI, as shown in Fig. 2d Native PAGE (now Fig. 3d), does not lead in the treatment conditions we use to an increase in epichaperomes, which is observed with PU-H71 (compare 1st column on the Native, V-PU to 2nd column, LS-V), and therefore does not re-wire the connectome. The Native PAGE shows LSI pre-treatment blocks subsequent re-wiring by PU-H71, thus supporting the proposed mode of action. We rephrased the text accordingly: "We confirmed this both biochemically and functionally by showing that, pharmacologically blocking HSP70/C70 recruitment into epichaperomes prior to PU-H71 treatment, inhibited deployment of redundant paths (Fig. 3d, see epichaperome levels between V-PU and LS-PU) and rendered MiaPaCa2 cells as vulnerable to PU-H71 as those with baseline hyperconnected PPI networks (Fig. 3e)." We hope this clarification eliminates any ambiguity.

7. Fig 3a: It is not clear how PPIs are generated? Is this an experiment done by authors or is this a superimposition of chaperomics on known PPI in literature. This must be clarified as the interpretations are very different!

R1.7. Response: We are sorry for any confusion caused by the previous description. Epichaperomics is an affinity purification method that uses the multitude of epichaperomes in a particular cell as bait to capture the disease-promoting complement of proteins and query their disease-specific interactions. An analogy is affinity purification methods whereby several tagged proteins are added to the cell and interacting proteins are captured. Another analogy is yeast two-hybrid (Y2H) experimentation. Unlike classical affinity purification and Y2H however, in epichaperomics there is no need to introduce artificial baits because the epichaperomes themselves are the baits. Also, in epichaperomics there is not one bait but rather a multitude of baits (i.e., the epichaperome oligomers characteristic of a specific cellular context), each having distinct interactors. Similar to other affinity purification methods, bait-bound interactors are captured and interrogated by mass spectrometry for identification. In the case of epichaperomics, the epichaperome 'biological' baits are captured by chemical probes that trap epichaperomes bound to their interactome (e.g., PU-beads), thus retaining interactions through subsequent isolation steps and enabling their unbiased identification by mass spectrometry. Similar to other conventional affinity purification methods, epichaperomics does not indicate which proteins are directly interacting with each other. Bioinformatics inference, not direct physical experiments, is used to generate proteome-wide interactome maps. In most cases, including for the purpose of this study, it is not important to know the direct binary interactions at the molecular or quantitative level, only which maladaptive proteins and pathways are stabilized which can be learned from the interactome data and bioinformatics, with further evidence based on functional assays and outcome assessments, as supported by our previous publications and as we also show here. To summarize, like with any affinity purification methods, PPIs are generated through experimentation (i.e., physical determination of binding interactors) combined with PPI database mining. For clarity, we added this paragraph along with supporting citations (page 7).

8. Page 7, line 26: The data cannot be interpreted that the pathways at baseline are supported by epichaperome - they are just bound to the oligomeric structure.

R1.8. Response: Reviewer #1 is correct indicating the standalone Cytoscape map does not show epichaperome dependence. Importantly, it is the combination of this experiment with biochemical and functional validation described further in the manuscript that support this statement. We revised the text to read, “Aberrant activity of these pathways at baseline is supported by epichaperomes (**Supplementary Fig. 3a** and see **further**)”.

9. Fig 3d: Is there anything special about proteins that are deployed/ pre-existent/ unchanged by pre-treatment? Any specific folds in proteins/ oligomeric structures/ ligand-dependence?

R1.9. Response: This is a very interesting comment. Indeed, much can be learned from studying the epichaperome composition as well as its interactors at ‘Baseline’ and ‘Rebound’. Our study here is limited in scope and evaluates mainly two outcomes. One is the nature of chaperones and co-chaperones recruited to the oligomeric structures at rebound, thus ‘deployed’ to execute and support the state of hyperconnectivity (Fig. 3). Another is the nature and identity of proteins co-opted at rebound, thus ‘deployed’ at rebound to support the activity of intrinsic signaling pathways. In response to this critique as well as a suggestion by Reviewer #2, we show proof-of-principle for the ‘proteins that are deployed by pre-treatment’, thus informing on alternate pathways that maintain RAS-MAPK signaling at rebound (Fig. 4c). See also response to Reviewer #2, R.2.11. and newly added Fig. 4c.

10. Fig 4a: Is this a blot of total proteins or PU-bound proteins? Almost all proteins look the same in T₀ and 1 μM. A quantification of enough replicates along with the representation of MS data for these proteins will be useful.

R1.10. Response: We specify in the text and the Figure Legend that is the “Western blot of effectors of signaling pathways constitutively active in MiaPaCa2 (baseline) and at recovery (24 h to 72 h after 1 μM PU-H71 wash-off)” showing the “activity of several pathways at baseline and at recovery (see Western blot analysis of pancreatic cancer signalling pathways at T₂₄₋₇₂ in Fig. 5a). The comparison is between T₀ (the time PU-H71 is removed) and T₂₄ to T₇₂ (the period of 3 days after PU-H71 is removed). At 1 μM PU-H71, note the activity of all signaling effectors (e.g., p-MEK, p-ERK, and p-RAF1, among others) recovers from the T₀ levels to baseline levels (i.e., levels seen in the 0 μM PU-H71 conditions). Conversely, after removal of the 5 μM PU-H71 (control condition) no signalling rebound and/or recovery is observed. We added the following paragraph to the text, “To show that cellular rebound is executed through reactivation of protein pathways already existent, and constitutively active at baseline in MiaPaCa2 yet deployed at recovery through alternate paths, we analysed the activity of signalling effectors (i.e. p-MEK, p-ERK, p-RAF1, p-AKT, p-S6, p-mTOR and p-STAT3) (**Fig. 5a,b**). We also tested the vulnerability of cells at recovery to inhibitors of these signalling pathways (see cell confluency and apoptosis in **Fig. 5c-e**). In MiaPaCa2 cells challenged with 1 μM PU-H71, these recovered from the T₀ levels (T₀, the time PU-H71 is removed from the culture) to baseline levels (i.e., levels seen in the 0 μM PU-H71 conditions at T₀ through T₇₂) in the period of 3 days (i.e., T₂₄ to T₇₂) monitored after removal of PU-H71 (**Fig. 5a**). Conversely, no signalling rebound was observed in cells challenged with 5 μM PU-H71 (**Fig. 5a**), the control condition of complete epichaperome suppression and thus of no rebound possible.” Page 10, lines 8-18.

11. Fig 4b: There is no difference in T48 cargo in 0/ 1 PU. It is not clear what the authors want to conclude from this.

R1.11. Response: As per **R1.10**, this result supports the notion that signaling at rebound occurs through reactivation of baseline signaling pathways and remains supported by epichaperomes. We added the following text in combination with the inquiry **R1.10**: “Supportive of a retained dependence of these rebound signalling effectors on epichaperomes is their capture by the PU-beads at both baseline and recovery (see Fig. 5b, compare beads cargo at T₄₈ in cells released from either 0 or 1 μ M PU-H71).”. page 10, lines 18-20.

12. In general the text is extremely difficult to read for a reader who has not followed all the previous work from the lab, and is not familiar with atypical terms such as maxed-out, hyperconnectivity etc. It is in authors' interest to explain their blots and interpretation in simpler terms/ more words. The first two figures are info-intense whereas the last few figures are much easier to follow. A redistribution of the data could be considered to simplify the paper's message. Similarly, the abstract does not convey the data but is written to mostly communicate interpretations of the authors. Specific molecular details replacing diffuse, unclear phrases, would be critical for revision of the abstract.

R1.12. Response: We have taken this critique seriously and have streamlined text throughout the manuscript as well as revised the Abstract, Figures, and Figure Legends for clarity. We've also defined some of the terms that may be novel to interested readers. Once again, thank you for requesting clarifications to the text and to redistribute data in the figures, which we did as indicated in **R1.1- R.1.11**.

Reviewer #2: Joshi et al. employ a chemical modulation approach of the cellular interactome to a hyperconnectivity state and show that this is associated with the increased response of pancreatic cancer cell lines to specific drugs including those that target the MAPK-pathways and PI3K-mTOR pathway. Furthermore, they show a proof-of-concept that this chemical modulation of the interactome achieved via epichaperome inhibition by the small molecule PU-H71 may hold great clinical prospects in treating pancreatic cancer and among other cancers. This is because priming interactomes as they showed may increase the vulnerability of cancer cell to drugs and thus improve drug efficacy. Overall, the methods and results of this paper are well done and straightforward to follow. I think that the current paper holds great biomedical significance, but at the same time, we can only realise such relevance upon further experiments and follow-through results as I have highlighted in the comments to the authors.

1. Page 7, line 2: The authors state that "Thus, analogous to baseline interactome hyperconnectivity, observed in MDA-MB-468 and other tumours, chemically induced hyperconnectivity is executed by enhancing the number of interactions and interaction partners between components of the two major chaperone networks, HSP90 and HSP70". Is the relative contribution of HSP90 and HSP70 toward the shift to the hyperconnectivity state the same across different cell lines of the same cancer types, and/or cancer of different origins? If not (which I suspect), in a follow-up study, the authors could investigate this across various cancer types, other cell lines and xenograft of each of the cancer types and show the impact of one HSP having a more significant influence toward chemically induced hyperconnectivity against the other. i.e., one of the HSP may be specific to particular tissues, or more effective than the other, which may also be depended on the tissue, which ultimately may affect the utility of this approach to treat tumours of specific origins.

R2.1. Response. Thank you for this insightful comment. Indeed, we believe there are several lines on inquiry of therapeutic significance that will be opened by this study for future consideration.

2. Page 8, line 15: “We chose FOR study MiaPaCa2 and HPAF-II cell lines, both characterized at baseline by medium epichaperome levels, and thus of partly occupied network capacity”. It appears that the preposition "FOR" in the sentence is incorrect. The authors should consider replacing it with "TO".

R2.2. Response. We corrected the sentence.

3. Page 8, line 21: Importantly, each of these inhibitors became highly toxic to cells in the state of engineered hyperconnectivity (Fig. 5b). The reverse order of addition, i.e., MEK or PI3K inhibitor added before 23 PU-H71, was less toxic (Fig. 5c). Did the authors investigate whether this effect is synergistic or additive by treating the cell with both PU-H71 and a MEK or PI3K inhibitor simultaneously? There is a possibility that a significant amount of PU-H71 may remain (even after washing) within the cell or bound to some cellular proteins. Maybe, alternatively (more straightforward), the authors could show that the washing step in their approach completely depletes the cancer cells of PU-H71. I should state that whether the effect is synergistic, or additive does not in any way invalidate the authors finding on the impact of PU-H71 treatment followed by a kinase inhibitor on xenografted tumours in mice.

R2.3. Response. We added a new figure, Fig. 7, to show the effect of simultaneous versus sequential addition of inhibitor. We believe this thoughtful suggestion adds value to this manuscript. The following text was added to accompany Figure 7, "To understand how the results from the sequential treatment compare to the classical simultaneous combination treatment, we performed both live-microscopy monitoring to evaluate cell recovery, or lack of (Fig. 7a,b), as well as full isobologram treatment of PU-H71 (0 to 1 μ M) and Trametinib (0 to 500 nM) to evaluate synergy (Fig. 7c). The simultaneous combination of PU-H71 and Trametinib was both cytotoxic and synergistic, which confirms the results of a recent study where it was identified as most active in a large-scale, unbiased *in vivo* screen of 57 different single agent and combination targeted therapies⁴⁴. However, we observed cell recovery was substantially delayed in the sequential treatment paradigm when compared to the simultaneous drug-addition regimen. This was evidenced in the cell confluency curves recorded over the time cells were kept in culture following drug removal (Fig. 7a) and through images of the cell culture plate taken at day 7 (Fig. 7b). Supportive of distinct mechanisms of drug action, the synergy map over the dose matrix was also different, with the epichaperome-mediated PPI hyperconnectivity map showing high synergistic effect across most evaluated drug concentrations, whereas the simultaneous drug regimen was sensitive to the concentration of each agent in the combination (Fig. 7c)." Page 11, lines 5-19.

4. In their previous publication (ref. 18) which the authors have cited several times in the current paper, they show that "under conditions of stress, such as malignant transformation fuelled by MYC, the chaperome becomes biochemically 'rewired' to form a network of stable, survival-facilitating, high-molecular-weight complexes." In the same publication, the authors show two types of cell lines, type-1 and type-2, associated with MYC expression and that PU-H71 binds strongly to HSP90 in the type-1 cell lines. Here, in the current publication, the authors should have carried out their experiments on both a type-1 cell line and a type-2 cell line as the type of cell line may be a factor that influences the clinical utility of their approach. Again, this is something that the authors can show in a follow-up study.

R2.4. Response. This comment provides another excellent suggestion for follow-up studies. Our approach is designed for type 1 tumours, notably tumours that have epichaperomes and thus are epichaperome-dependent. As noted in the Introduction, "Independent of tissue of origin, tumour subtype, or genetic background, approximately 50–60% of tumours express variable epichaperome levels", a percentage confirmed here also in primary pancreatic specimens and pancreatic cancer cells (Fig. 1a,b), and we surmise this is the number of tumours amenable to this potentially novel treatment approach. Certainly, this will have to be demonstrated across tumour types, rigorously in cultured cells lines, *in vivo* tumours and primary specimens, a large undertaking outside the scope of this manuscript which demonstrates proof-of-principle. A sentence was added to Discussions, page 13.

5. Page 8, line 29, the authors state that "Trametinib had little to no activity in naïve ('baseline') tumours, as previously reported²⁸. Poor efficacy of Trametinib and other MEK inhibitors in pancreatic cancer is caused by a reciprocal increase or maintenance in alternate signalling activity, such as the AKT/mTOR pathway activity. We observed the deployment of such alternate paths was suppressed when tumours were forced into the interactome hyperconnectivity state." This is the main punchline of the study as many studies show that a significant number of tumours are resistant to drugs because of alternative pathways being activated that lead to the activation of the same downstream transcription factors and mechanisms that the drug should abate. The "alternative signalling activity" and how the authors' approach takes

care of this conundrum in cancer cells should even be mentioned in the abstract, if possible, as there is space for a few more words in their abstract.

R2.5. Response. We thank the Reviewer for this interpretation of our work. We have revised the Abstract and the text to better reflect the concept of "alternative signalling activity".

6. About the title "Interactome engineering for therapeutic vulnerability in cancer". I believe the title and to a lesser extent the abstract of the study undersell the importance of the authors' findings. I don't have a better title for the manuscript, but the journal Editor and the authors could develop a title that will give more prominence to the paper. This title should have some information on pharmaceutical modulation (not engineering) of cellular networks to a hyperconnectivity state leading to increased drug sensitivity. Furthermore, I think that "interactome engineering" part of the title may be somehow misleading (a riddle or brainteaser) that might undersell this paper's importance. Possibly a few simple and direct words would increase the visibility and readership of this great work.

R2.6. Response. In response to this critique, we revised the title to "Pharmacologically controlling protein-protein interactions through epichaperomes for therapeutic vulnerability in cancer".

7. The discussion section also starts with a "brainteaser" sentence: "We demonstrate the discovery of an approach for engineering the interactome into a state of hyperconnectivity and maxed-out capacity for therapeutic vulnerability in cancer and present experimental evidence for interactome capacity engineering.". I wish that the authors could write these sentences using simple and straight forward words like the two sentences that follow the opening statement of the discussion: "This achievement, with proof-of-principle provided here in pancreatic cancer, sets the stage for future efforts aiming to implement this strategy to other drugs and tumour types. Such knowledge may open the door for innovative, and feasible, cancer therapies based on understanding the capacity of interactome networks rather than trying to correct a pathway defect that has redundancy that lends itself to being refractory to drug treatments."

R2.7. Response. Similar to the response to a comment of Reviewer #1 (**R1.12**), we have made a concerted effort to streamline and simplify the text from the Abstract to the Conclusions in order to present our results and interpretations more clearly and succinctly.

8. I agree with the authors' statement that "we believe this approach will be a landmark method to engage PPI networks for therapeutic intervention by maxing out hyperconnectivity". However, there should be follow-through studies to provide evidence across various human cancer types and cancer subtypes, those that express higher or lower MYC levels (i.e., type-1 and type-2), or tumours that have mutations in genes or express higher levels of specific proteins that may affect the interactome priming. The authors could also investigate how the primed cells respond to drugs that target other pathways (other than the MAPK and PI3K pathways) for which there is likely to be variations in drug efficacy.

R2.8. Response. These are excellent suggestions for rigorous follow-up studies. We added a comment to this effect to the Discussion section, "This suggests 50-60% of all tumours, independent of tumour type, could be engineered for interactome hyperconnectivity. Follow-through studies to provide evidence across various human cancer types and cancer subtypes, those that express higher or lower epichaperomes, or tumours with mutations in genes or expressing higher levels of specific proteins that may affect the interactome priming will be needed to provide further mechanistic and therapeutic insights." Page 13, lines 12-17.

9. The title says, "in cancer", but the authors only show a proof-of-concept for pancreatic cancer. Maybe the word cancer could be return for the sake of having a broader readership for the paper.

R2.9. Response. We have revised the title for the broader implications of the findings in cancer that also fit with the goals and perspectives of *Communications Biology*. As stated in **R2.6** the revised title is "Pharmacologically controlling protein-protein interactions through epichaperomes for therapeutic vulnerability in cancer".

10. The protein-protein interaction network changes observed after chemical perturbation are likely linked to changes in transcription factor expression and activity. Can the author provide information on transcription factor enrichments (the Ma'ayan Lab's Enrichr tool can do this) between the baseline and hyperconnectivity state of the MiaPaCa cell line? Also, they could analyse for the enriched transcription factors between the baseline MiaPaCa cell line and the MDA-MB-468 cell line (which is already in a hyperconnectivity state) and see if the same transcription factors show up in the comparisons of the baseline MiaPaCa vs primed MiaPaCa and the baseline MiaPaCa vs MDA-MB-468.

R2.10. Response. As suggested, we investigated transcription factor enrichments between the baseline and the hyperconnectivity state of the MiaPaCa2 cell line. Results are added to Supplementary Fig. 3b (see also pasted here on the right). See also R2.11 how these analyses support our findings of how 'new signalling routes are activated within several signalling pathways'. This text was added to page 9: "Gene set enrichment analysis of transcriptional factors differentially expressed between MiaPaCa2-PU and MiaPaCa2 support enhanced NFkB transcriptional activity at recovery (Supplementary Fig. 3b)."

11. The results shown in Figure 3d are quite remarkable! The authors show that new signalling routes are activated within several signalling pathways, including the MAPK pathway, PI3K pathway, TGF-β pathway and among others. It would be nice if the authors could also evaluate the effect of treating the cancer cell in the hyperconnectivity state with 1) a drug that targets a protein (e.g., a kinase) within the 'existent' signalling route, 2) a drug that targets a protein in the 'deployed' signalling route, and 3) a combination of both drugs. Furthermore, it would be nice if the authors could evaluate the effect of treating the unprimed (baseline) MiaPaCa cell line with a combination of drugs that target proteins within both the 'existent' and 'deployed' signalling route of a signalling pathway of their choosing. If this approach works similar to the priming followed by treatment approach, then the priming approach could be used to create a knowledgebase of information on 'existent' and 'deployed' in cancer and thus specific drug combinations that are likely more effective at treating particular cancer types.

R2.11. Response. These are excellent points, and clearly the Reviewer understands the potential power and importance of these studies for treatment regimens. A caveat in performing the aforementioned experiments is that truly specific and well-validated pharmacologic agents to target many discrete proteins within critical signalling

pathways are yet to be developed. Further, knock-down experiments are incompatible with dynamic protein network studies.

Consequently, and for the purpose of this manuscript, we pharmacologically explore the 'deployed' signaling route as

a proof-of-principle. Specifically, we chose the Ras-MAPK signaling to inquire on proteins utilized by MiaPaCa2 cells to maintain the activity of this pathway at rebound (see heatmap in Fig. 4c). We identified that the 'deployed' MAPK signaling output is executed through an interplay of pathways such as NFkB activation by PKA and by RAF activation by PLCγ. Enhanced NFkB transcriptional activity at recovery was also supported by gene set enrichment analysis of transcriptional factors differentially expressed between MiaPaCa2-PU and MiaPaCa2 (Supplementary Fig. 3b, and see R2.10). We then used established pharmacologic agents targeting key proteins in the deployed state. These are reported to act on inhibiting NFkB transcription (JSH-23), PLC activity (U-73122) and cAMP-dependent protein kinase (PKA) (PKA inhibitor fragment (6-22) amide and the small molecule H-89). Where available, structurally similar inactive controls were also included (e.g., U-73343). SCH772984, an

ERK1/2 inhibitor was also included as control to confirm ERK activation, as a read-out for Ras-MAPK signaling activity in the 'deployed' state. We found these pharmacologic inhibitors lowered signaling output in the deployed state as evidenced by a reduction in p-ERK levels, thus supporting the notion that "the priming approach could be used to create a knowledgebase of information on 'existent' and 'deployed' connections in cancer" for informing on potential drug combinations that are more likely to be effective at treating particular cancers. A summary of this information was added to the Results and the Discussion sections (p. 9,10 and p. 12).

12. Tissue or cell-type-specific variations in the wiring of cellular networks are attributable to the differences in the expression profile of proteins – specific proteins are not expressed in some tissues, mostly because of epigenetics. Therefore, it is improbable that the cell lines can shift from the baseline state to the hyperconnectivity state without changes in the epigenome, e.g., the proteins that were not previously expressed (in the baseline state) but required to link two or more proteins in a network must be present (in the hyperconnectivity state). As such, I suggest that the authors explore changes that occur in the epigenome in cancer cells between the baseline and the hyperconnectivity state. Such information would give us more valuable insights on how to treat cancers.

R2.12. Response. Thank you for another excellent comment. Connectivity changes detected by epichaperomics are changes that occur in the epigenome. These PPI changes effected by aberrant scaffolding platforms (i.e., epichaperomes) occur with or without PPI-participant proteins changing their expression level as seen in our previous publications, and we are actively investigating these complex mechanisms in the context of cancer as well as in neurodegenerative disorders for breadth and depth of understanding the pathobiology of epichaperome formation and its therapeutic implications.

REVIEWERS' COMMENTS:

Reviewer #1 (Remarks to the Author):

The authors have addressed all major concerns raised in the review. I would only recommend them to further tone down the text to make the paper understandable by a larger audience - in line with comments that both reviewers have made.

Reviewer #2 (Remarks to the Author):

The authors have addressed all my concerns in the current version of the manuscript. I have enjoyed reading this manuscript and look forward to the authors' future work.

We thank the Reviewers for their thoughtful and constructive critiques throughout the review process. We are pleased they concluded this manuscript is ready for publication in *Communications Biology*.

Below we address in point-by-point fashion the remaining comments of the Reviewers.

Reviewer #1 (Remarks to the Author):

The authors have addressed all major concerns raised in the review. I would only recommend them to further tone down the text to make the paper understandable by a larger audience - in line with comments that both reviewers have made.

Response: Thank you for appreciating our diligence in responding to the critiques. We have taken the suggestion to make the paper understandable by a larger audience seriously throughout the review process. In line with comments made by both reviewers, we have streamlined text throughout the manuscript. Revisions included defining novel terms, clarifying the description of experiments, results, and their interpretations and implications. We also included expanded schematics in the figures to illuminate the background, concept and experimental design. We could not expand the Abstract due to word limitations. Rather, we highlighted specific key findings in the final paragraphs of the Introduction.

Reviewer #2 (Remarks to the Author):

The authors have addressed all my concerns in the current version of the manuscript. I have enjoyed reading this manuscript and look forward to the authors' future work.

Response: Thank you – this is very kind of Reviewer #2 and we are grateful for her/his input.